# Motor cortex analogue neurons in songbirds utilize Kv3 channels to generate ultranarrow spikes

Benjamin M Zemel[1], Alexander A Nevue[2], Leonardo ES Tavares[1,3], Andre Dagostin[1], Peter V Lovell[2], Dezhe Z Jin[3], Claudio V Mello[2]*, Henrique von Gersdorff[1,4]*

[1]Vollum Institute, Oregon Health and Science University, Portland, United States; [2]Department of Behavioral Neuroscience, Oregon Health and Science University, Portland, United States; [3]Department of Physics, Pennsylvania State University, University Park, United States; [4]Oregon Hearing Research Center, Oregon Health and Science University, Portland, United States

*For correspondence:
melloc@ohsu.edu (CVM);
vongersd@ohsu.edu (HvG)

**Abstract:** Complex motor skills in vertebrates require specialized upper motor neurons with precise action potential (AP) firing. To examine how diverse populations of upper motor neurons subserve distinct functions and the specific repertoire of ion channels involved, we conducted a thorough study of the excitability of upper motor neurons controlling somatic motor function in the zebra finch. We found that robustus arcopallialis projection neurons (RAPNs), key command neurons for song production, exhibit ultranarrow spikes and higher firing rates compared to neurons controlling non-vocal somatic motor functions (dorsal intermediate arcopallium [AId] neurons). Pharmacological and molecular data indicate that this striking difference is associated with the higher expression in RAPNs of high threshold, fast-activating voltage-gated Kv3 channels, that likely contain Kv3.1 (*KCNC1*) subunits. The spike waveform and Kv3.1 expression in RAPNs mirror properties of Betz cells, specialized upper motor neurons involved in fine digit control in humans and other primates but absent in rodents. Our study thus provides evidence that songbirds and primates have convergently evolved the use of Kv3.1 to ensure precise, rapid AP firing in upper motor neurons controlling fast and complex motor skills.

## Editor's evaluation

This paper carries significant novelty, as it describes a mechanism for fast information processing in the innovative zebra finch model and provides strong experimental support for the fundamental cellular properties underlying such high-frequency signaling.

## Introduction

Zebra finches are an accessible model organism for studying how upper motor neurons control a rapid and precise learned behavior. Their song consists of bouts of repeated motifs made up of multiple syllables, each containing distinct features that change on a rapid timescale (*Leonardo and Fee, 2005*; *Sturdy et al., 1999*; *Yazaki-Sugiyama et al., 2015*). In sharp contrast with mammals, which possess a six layered neocortex, songbirds orchestrate song behavior with a circuitry consisting of pallial nuclei, which nonetheless have connectivity analogous to mammalian cortical microcircuits (*Jarvis, 2019*; *Karten, 2015*). Projection neurons within the robust nucleus of the arcopallium (RAPNs) represent the main forebrain output of the vocal motor system of songbirds. They share several molecular markers

with mammalian layer 5 pyramidal neurons (L5PNs) (*Dugas-Ford et al., 2012*; *Nevue et al., 2020*; *Pfenning et al., 2014*; *Stacho et al., 2020*). During song production, RAPNs exhibit robust burst-pause firing with high temporal precision (~0.2 ms variance in the first spike latency; *Chi and Margoliash, 2001*). They are thus well poised to orchestrate the rapid movements of the syringeal muscles (~4 ms to peak force of muscle twitch; *Adam and Elemans, 2020*).

We have recently shown that zebra finch RAPNs share many electrophysiological properties with mammalian L5PNs, including a hyperpolarization-activated current ($I_h$), a persistent Na$^+$ current, and a large transient Na$^+$ current ($I_{NaT}$) with a short onset latency (*Zemel et al., 2021*). These properties facilitate the minimally adapting, regular spiking of APs (*Almog et al., 2018*; *McCormick et al., 1985*). A major difference, however, is that RAPNs exhibit an ultranarrow AP half-width (*Zemel et al., 2021*), a property also found in primate Betz cells (*Lemon et al., 2021*; *Vigneswaran et al., 2011*). Not found in rodents (*Lacey et al., 2014*; *Soares et al., 2017*), these cells are sparsely distributed across layer 5 of the motor cortex in primates and cats (*Chen et al., 1996*; *Rivara et al., 2003*; *Tomasevic et al., 2022*), often terminate directly onto lower motor neurons and are thought to facilitate highly refined aspects of motor control (*Lemon and Kraskov, 2019*). The unique AP half-width of Betz cells is associated with high expression of the voltage-gated potassium channel Kv3.1 subunit (*Bakken et al., 2021*; *Ichinohe et al., 2004*; *Soares et al., 2017*). The high-voltage, fast activation and fast deactivation of these channels have been shown to significantly narrow the AP waveform, thus facilitating high-frequency firing (*Hong et al., 2016*; *Kaczmarek and Zhang, 2017*; *Rudy and McBain, 2001*).

Similar to mammalian Betz cells, RAPNs in adult male finches also exhibit high expression of *KCNC1* (Kv3.1; *Lovell et al., 2013*). Interestingly, this expression is significantly lower in the adjacent dorsal intermediate arcopallium (AId) (*Nevue et al., 2020*), an area thought to be involved in non-vocal somatic motor functions (*Feenders et al., 2008*; *Mandelblat-Cerf et al., 2014*; *Yuan and Bottjer, 2020*) that have different temporal requirements than song (e.g., ~10 ms to peak force of twitch for pectoral wing muscles *Bahlman et al., 2020*). Whereas the excitable properties of RAPNs have been examined in detail (*Adret and Margoliash, 2002*; *Garst-Orozco et al., 2014*; *Liao et al., 2011*; *Spiro et al., 1999*; *Zemel et al., 2021*), little is known about (1) the excitability of AId neurons and (2) the molecular basis shaping the AP waveform in either brain region.

A comparison of excitable features between RAPNs and AId neurons may reveal molecular specializations that evolved to specifically support the execution of complex, learned vocalizations. Here, we show that compared to RAPNs, AId neurons have broader APs and spike at lower frequencies during current injections. We found that these differences are due, in part, to the differential activity of Shaw-related K$^+$ channels (Kv3.1–3.4), as Kv3 channel blockers disproportionately broadened APs and decreased firing rates in RAPNs compared to AId neurons. Furthermore, a novel Kv3.1/3.2 positive modulator, AUT5, narrowed the AP waveform and increased the steady-state firing frequency of RAPNs, but not of AId neurons. Moreover, *KCNC1* had significantly higher expression in RA compared to AId, while *KCNC2–4* (Kv3.2–3.4) genes were non-differential in expression. Notably, our analysis identified zebra finch *KCNC3* (Kv3.3), a gene previously thought to be absent in birds. Morphological analysis revealed higher dendritic complexity and spine density but smaller spines in AId neurons compared to RAPNs, which had larger spines, like Betz cells (*Kaiserman-Abramof and Peters, 1972*). We propose that the shared molecular and electrophysiological properties that promote ultranarrow spikes in songbird RAPNs and primate Betz cells likely originated from a convergent evolutionary process that allowed these neurons to operate as fast and precise signaling devices for fine motor control.

## Results

Understanding the neuronal basis of specific behaviors requires determining the cellular properties of the upper motor neurons involved in those behaviors. RAPNs and AId neurons are thought to control vocal and non-vocal somatic functions respectively via descending projections out of the finch telencephalon toward lower motor centers (*Bottjer et al., 2000*; *Feenders et al., 2008*; *Mandelblat-Cerf et al., 2014*; *Nottebohm et al., 1976*; *Paton et al., 1981*; *Wild, 1993b*; *Yuan and Bottjer, 2020*; *Mello et al., 2019*; *Figure 1A*). Contrasting the properties of RAPNs and AId neurons may reveal specific features of vocal motor neurons (RAPNs) that have not been previously identified. RAPNs and AId neurons are known to differ in their molecular profiles, including higher expression of the voltage-gated potassium channel *KCNC1*/Kv3.1 in RAPNs (*Nevue et al., 2020*).

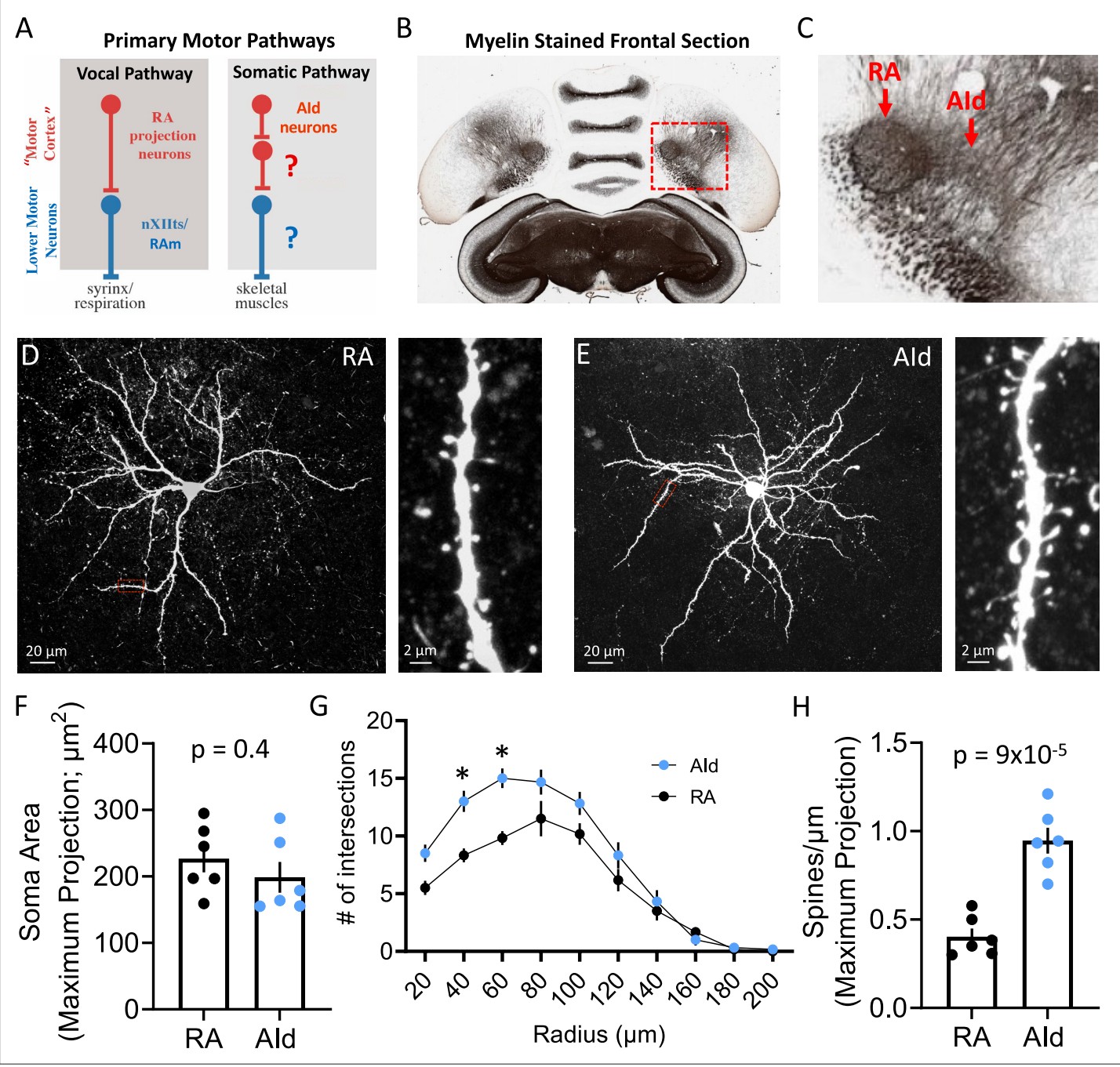

**Figure 1.** Morphology of robustus arcopallialis projection neurons (RAPNs) and dorsal intermediate arcopallium (AId) neurons. (**A**) Diagram of the motor pathways in the zebra finch. RA and AId upper motor neurons in the arcopallium project to downstream targets that innervate syringeal and skeletal muscles respectively. Note that the exact cellular projection targets for AId neurons in the brainstem (***Bottjer et al., 2000***) and/or spinal cord have yet to be determined. (**B**) Myelin-stained frontal section of a zebra finch brain, including RA and AId. Red box in (**B**) indicates the region containing RA and AId. (**C**) Expanded region shown in (**B**). Note the clear RA and AId arcopallial regions with inputs from the nidopallium above. Panels B and C are adapted from a myelin-stained frontal section of a zebra finch brain taken from the Brain Architecture Project (http://www.brainarchitecture.org/) ***Karten et al., 2013***, published under CC-BY-SA. (**D**) Left: Maximum projection of a Z-stack from a biocytin-filled RAPN. Red box indicates dendritic region expanded on the right. Right: Expanded region showing a 30 μm long dendritic section from the neuron to the left. (**E**) Left: Maximum projection of a Z-stack from a biocytin-filled AId neuron. Red box indicates dendritic region expanded on the right. Right: Expanded region showing a 30 μm long dendritic section from the neuron to the left. (**F**) Dot plot comparing the soma area of RAPNs and AId neurons. Student's t-test, two-tailed, $t_{stat}$ = 0.91, N=6 RAPNs and 6 AId neurons. Black bars indicate standard error. (**G**) Comparison of the dendritic complexity from RAPNs and AId neurons as determined by the number of intersections with concentric circles of increasing radii surrounding the cell body. Two-way ANOVA; p=0.004, F(9,

*Figure 1 continued on next page*

Figure 1 continued

100)=2.92, N=6 RAPNs and 6 AId neurons. *p<0.05 as determined by Tukey's post hoc test. Black bars indicate standard error. (**H**) Dot plot comparing the number of spines per micron of RAPNs and AId neurons. Spines were counted on a 30 µm dendritic segment after the second branch point. Between 2 and 4 segments were measured per cell, averaged and divided by 30 to determine the # spines/µm. Student's t-test, two-tailed, $t_{stat}$ = 6.29, N=6 RAPNs and 6 AId neurons. Black bars indicate standard error.

© 2008, Karten et al. Panels B and C are adapted from a Myelin-stained frontal section of a zebra finch brain taken from the Brain Architecture Project (http://www.brainarchitecture.org/) Karten et al. (2008), published under CC-BY-SA.

The online version of this article includes the following figure supplement(s) for figure 1:

**Figure supplement 1.** Reconstruction of neuron morphology and estimation of spine density.

Our goal here was to contrast the excitable properties of RAPNs and AId neurons in order to (1) identify and (2) determine the role of molecular correlates of excitability that differ between these neuronal populations.

## Morphological features of RAPNs and AId neurons

We started with an examination of the morphology of principal neuronal cells in RA and AId, whose close proximity in the arcopallium can be easily observed in frontal sections (*Figure 1B–C*; *Figure 2A–B*). Whereas many morphological characteristics of RAPNs have been described (*Hayase et al., 2018*; *Kittelberger and Mooney, 1999*; *Miller et al., 2017*; *Spiro et al., 1999*), those of AId neurons have not been previously examined. Intracellular biocytin fills in frontal slices revealed that recorded AId neurons exhibit several morphological characteristics similar to those of RAPNs, including a large soma, a large diameter axon initial segment with extensively branched thin, aspinous axonal collaterals, and an extensive dendritic arborization, with individual dendrites exhibiting numerous spines (*Figure 1D–F*). There were however some important differences. A Sholl analysis (*Sholl, 1956a*) revealed the extent of dendritic complexity for AId neurons to be significantly greater than that of RAPNs, although dendrites of AId neurons were restricted to ~200 µm from the cell body, as has been previously reported for RAPNs (*Figure 1G*). AId neurons also had approximately twice the number of spines compared to RAPNs as measured from a 30 µm segment of the tertiary branch of the filled cells (mean ± SE: 0.9±0.07 vs 0.4±0.05 spines/µm; *Figure 1H*). Additionally, whereas RAPN spines consistently presented with a mushroom-type shape, those in AId neurons exhibited a mix of mushroom, thin and filopodic structures (*Figure 1D–E*, right; for a review on spine shapes see *Pchitskaya and Bezprozvanny, 2020*).

To gain further insights into the morphology of RAPNs and AId neurons, we next analyzed our highest quality images (three RAPNs and two AId neurons) using ShuTu software designed for digital 3D reconstructions (*Figure 1—figure supplement 1A–B*; *Jin et al., 2019*). We were able to confirm the approximately twofold larger spine density when pooling all branch segments together (*Figure 1—figure supplement 1C*; *Figure 1*, *Supplementary file 1*). The distribution of these spines as a function of distance from the soma was unimodal in both groups, with an increase in spines occurring after the initial, mostly aspinous primary dendrite, followed by a decrease toward more distal branches (*Figure 1—figure supplement 1E*). Upon manually tracing >1000 individual spines for each group we found that AId neurons display a larger proportion of spines with smaller 2D surface areas compared to RAPNs (*Figure 1—figure supplement 1D*). The total surface area of the soma and dendrites of RAPNs and AId neurons is shown in *Figure 1*, *Supplementary file 1*. The average surface area for different compartments were (in µm²) soma: 879.7 vs 727; dendrites: 5047.7 vs 5548; spines: 638.7 and 937, for RAPN and AId neurons, respectively (axons not included; *Figure 1*, *Supplementary file 1*). While we acknowledge the limited data set and the caveat that some dendrites are cut off by the slicing procedure, the results suggest that AId neurons may receive more numerous, perhaps diverse, synaptic inputs that are differentially filtered compared to RAPNs. Finally, we calculated the surface to volume ratio. For RAPNs we find 1.5–2.3 µm⁻¹ and for AId neurons 2.5–2.9 µm⁻¹ (*Figure 1*, *Supplementary file 1*). These values are similar to those found using EM 3D reconstructions in mouse cortical neurons (1.63 µm⁻¹) but lower than found in astrocytes (4.4 µm⁻¹ *Calì et al., 2019*), which may have metabolic energy consequences for RA nuclei, which stain heavily for metabolic markers (*Adret and Margoliash, 2002*).

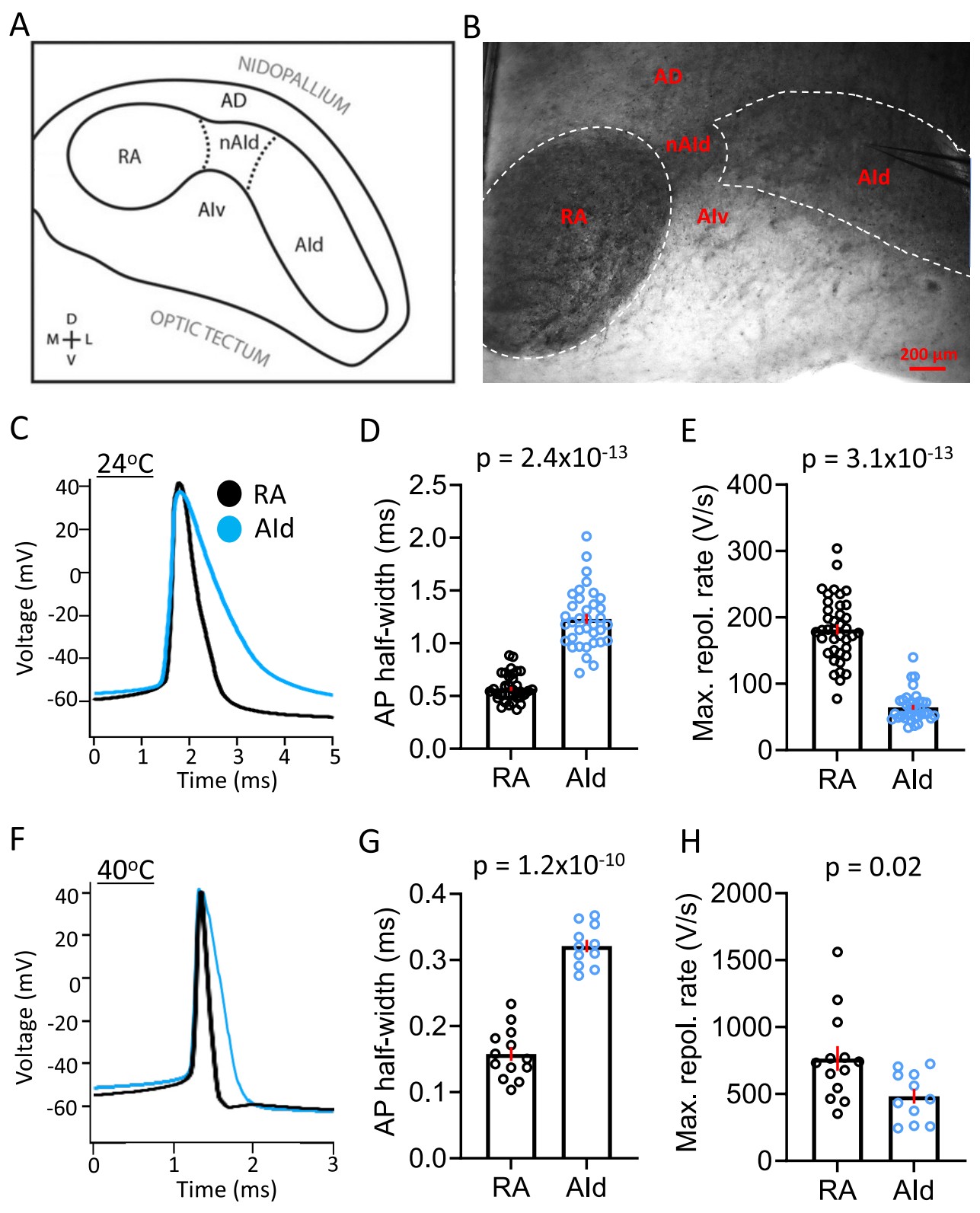

**Figure 2.** Spontaneous action potential (AP) properties in robustus arcopallialis projection neurons (RAPNs) and AId neurons from adult male zebra finches at room and physiological temperatures. (**A**) Drawing of the zebra finch arcopallium in the frontal plane, depicting RA and AId as defined by SCN3B staining in adult males. Labeling of other domains are included as referential landmarks. *AD*, dorsal arcopallium, *AId*, dorsal intermediate arcopallium, *AIv*, ventral intermediate arcopallium, *nAId*, neck of the dorsal intermediate arcopallium, *RA*, robust nucleus of the arcopallium. ***Figure 2A***

*Figure 2 continued on next page*

*Figure 2 continued*

is reproduced from Figure 2 from *Nevue et al., 2020*. (**B**) Infra-red differential interference contrast microscopy (IR-DIC) image of a frontal brain slice in which RA and AId are clearly visible, noting that the black elements seen correspond to heavily myelinated fibers. Labels correspond to those depicted in (**A**). Note the recording electrode in AId. (**C**) Overlay of averaged spontaneous APs recorded at ~24°C from RAPN (black) and AId neurons (blue). (**D**) Comparison of the average spontaneous AP half-widths from RAPNs and AId neurons recorded at ~24°C (Mann-Whitney U=11, two-tailed, N=39 RAPNs and 36 AId neurons. Red bars indicate standard error). (**E**) Comparison of the average spontaneous AP maximum repolarization rate from RAPNs and AId neurons recorded at ~24°C. Mann-Whitney U=14, two-tailed N=39 RAPNs and 36 AId neurons. Red bars indicate standard error. (**F**) Overlay of averaged spontaneous APs recorded at ~40°C from RAPN (black) and AId neurons (blue) respectively. (**G**) Comparison of the average spontaneous AP half-widths from RAPNs and AId neurons recorded at ~40°C. Student's t-test, two-tailed, $t_{stat}$ = 11.33, N=13 RAPNs and 11 AId neurons. Red bars indicate standard error. (**H**) Comparison of the average spontaneous AP maximum repolarization rate from RAPNs and AId neurons recorded at ~40°C. Student's t-test, two-tailed, $t_{stat}$ = 2.50, N=13 RAPNs and 11 AId neurons. Red bars indicate standard error. p-Values are included in the graphs.

The online version of this article includes the following figure supplement(s) for figure 2:

**Figure supplement 1.** Interneurons vs. projection neurons in dorsal intermediate arcopallium (AId).

## RAPNs and AId neurons: passive properties

We chose the frontal plane of sectioning for our recordings, as it facilitates the identification of both RA and AId on the same slice (*Figure 2A–B*), noting the heavy myelination of the arcopallial area containing RA (*Champoux et al., 2021*) and AId (*Figures 1B–C , and 2B*). A comparison of RAPNs and AId neurons recorded in frontal sections at both room temperature (24°C) and physiologically relevant temperature (40°C, *Aronov and Fee, 2012*) in the whole-cell current-clamp configuration revealed only minor differences in several passive membrane properties, including the membrane time constant, input resistance, and the calculated membrane capacitance ($C_m$) (*Table 1*). Using the calculated $C_m$ from our room temperature recordings, we determined that the estimated average surface area was not significantly different between RAPNs and AId neurons (mean ± SEM: 11,563.6±2277.5 µm² vs. 7578.6±1564.7 µm² for RAPNs and AId neurons, respectively; Student's t-test, $t_{stat}$ = 1.331, p=0.2). When considering possible tissue shrinkage (≤20%) during processing for fluorescent imaging (*Winsor, 1994*), these values approximate those found from results obtained from ShuTu total surface area reconstructions (adjusted mean: 8207.5 µm² vs 9031.9 µm² (including the soma, dendrites, and spines) for RAPNs and AId neurons, respectively; *Figure 1*, *Supplementary file 1*).

## RAPNs fire ultranarrow spikes at higher frequencies than AId neurons

RAPNs produce remarkably narrow APs (half-width = ~0.2 ms at 40°C; *Zemel et al., 2021*), making them well suited for orchestrating rapid movements of the syringeal and respiratory muscles required for song production. Upon examining properties of spontaneous APs, we found that RAPNs and AId neurons shared similar threshold, amplitude, peak, and after-hyperpolarization at both temperatures examined (*Figure 2C*; *Table 1*). However, AId neurons had an AP half-width that was twice as broad as the RAPN APs (*Figure 2C–D , and F–G*; *Table 1*). The shorter half-width of RAPNs could be due to either a faster AP depolarization and/or repolarization rate. To determine which phase of the AP was responsible for this difference, we derived phase plane plots from averaged spontaneous APs and compared the maximum rates of depolarization and repolarization. At room temperature, the maximum rate of repolarization was 76% larger in RAPNs compared to AId neurons (mean ± SEM in V/s: 181.9±7.8 vs 64.5±3.7, respectively; *Figure 2E*; *Table 1*), whereas the maximum rate of depolarization was only 29% larger in RAPNs compared to AId neurons (mean ± SEM in V/s: 658.4±26.5 vs 477.1±24.6, respectively; *Table 1*). At 40°C the difference in the maximum depolarization rate was not significant, while the maximum repolarization rate was still 37% greater in RAPNs (mean ± SEM in V/s: 764.7±92.4 vs 482.3±52.8, respectively; *Figure 2H*; *Table 1*). While the similar maximum rate of depolarization at 40°C may result from limitations on temporal resolution (*Oláh et al., 2021*), taken together with the room temperature recordings these findings indicate that RAPN APs are narrower than those of AId neurons, with larger differences in the maximum repolarization rate compared to the depolarization rate.

We next examined the evoked firing properties by delivering increasing, step-wise 1 s current injections to RAPNs and AId neurons (*Figure 3A and C*, examples at +500 pA). As expected, a higher firing rate was observed in both brain regions at physiological temperature compared to room temperature. In the absence of injected current, both RAPNs and AId neurons fired spontaneous APs (spontaneous APs in RAPNs also observed in extracellular slice recordings; *Wood et al., 2011*), however, RAPNs

**Table 1.** Passive and spontaneously active spike properties of adult male robustus arcopallialis projection neurons (RAPNs) and dorsal intermediate arcopallium (AId) neurons.

Statistical analysis compared RAPN and AId neurons at each temperature respectively. Statistics were performed using unpaired, two-tailed, Student's t-tests if data was normally distributed with equal variances. Otherwise a Mann-Whitney U test was used. Data shown as mean ± SEM. Room temperature comparisons: Spont. AP frequency - p=0.7, $t_{stat}$ = 0.3825, AP thresh - p=0.09, $t_{stat}$ = 1.696, AP amp - p=0.05, $t_{stat}$ = 1.971, AP half-width - p=$2.4 \times 10^{-13}$, U=11, Max. depol. rate - p=$4.0 \times 10^{-6}$, $t_{stat}$ = 4.988, Max. repol. rate - p=$3.1 \times 10^{-13}$, U=14, AP peak - p=0.2, $t_{stat}$ = 1.255, AHP - p=0.0001, $t_{stat}$ = 4.090, $\tau_m$ - p=0.5, $t_{stat}$ = 0.7259, $C_m$ - p=0.2, $t_{stat}$ = 1.326, $R_{in}$ - p=0.3, $t_{stat}$ = 1.160. High-temperature comparisons: Spont. AP frequency - p=0.0002, U=14, AP thresh- p=0.7, $t_{stat}$ = 0.3301, AP amp - p=0.2, $t_{stat}$ = 1.302, AP half-width - p=$1.2 \times 10^{-10}$, $t_{stat}$ = 11.33, Max. depol. rate - p=0.6, $t_{stat}$ = 0.4644, Max. repol. rate - p=0.02, $t_{stat}$ = 2.504, AP peak - p=0.1, $t_{stat}$ = 1.568, AHP - p=0.006, $t_{stat}$ = 3.025, $\tau_m$ - p=0.01, $t_{stat}$ = 2.735, $C_m$ - p=0.3, $t_{stat}$ = 1.027, $R_{in}$ - p=0.004, $t_{stat}$ = 3.280. Statistical significance: *=p < 0.05.

| | RA (24°C) | AId (24°C) | RA (40°C) | AId (40°C) |
|---|---|---|---|---|
| Spont. AP freq. (Hz) | 13.6±1.71 N=39 | 12.9±1.14 N=36 | 41±6.70 N=13 | 13.2±2.47 N=11 |
| AP thresh. (mV) | −55.3±0.66 N=39 | −53.7±0.62 N=36 | −52.0±1.00 N=13 | −52.7±1.71 N=11 |
| AP amp. (mV) | 92.8±1.53 N=39 | 89.0±1.39 N=36 | 96.3±3.57 N=13 | 102.8±3.28 N=11 |
| AP half-width (ms) | 0.57±0.02* N=39 | 1.23±0.05 N=36 | 0.16±0.01* N=13 | 0.32±0.01 N=11 |
| Max. depol. rate (V/s) | 658.4±26.53* N=39 | 477.1±24.59 N=36 | 1288.5±67.11 N=13 | 1339.4±85.34 N=11 |
| Max. repol. rate (V/s) | 181.9±7.75* N=39 | 64.5±3.71 N=36 | 764.7±92.41* N=13 | 482.3±52.83 N=11 |
| AP peak (mV) | 37.4±0.96 N=39 | 35.5±1.19 N=36 | 44.5±2.87 N=13 | 50.3±2.09 N=11 |
| AHP (mV) | −71.9±0.53 N=39 | −68.4±0.67 N=36 | −65.7±0.92 N=13 | −69.2±0.61 N=11 |
| $\tau_m$ (ms) | 17.4±1.9 N=8 | 15.2±2.3 N=6 | 17.2±1.36* N=10 | 13.3±0.60 N=12 |
| $C_m$ (pF) | 115.4±22.77 N=8 | 75.8±15.65 N=6 | 96.1±7.39 N=10 | 106.8±7.44 N=12 |
| $R_{in}$ (MΩ) | 176.6±22.11 N=8 | 209.2±13.03 N=6 | 185.2±15.02* N=10 | 129.08±8.81 N=12 |

Abbreviations - Spont. AP frequency: AP spikes produced per second, AP thresh.: AP threshold, AP amp.: AP amplitude as measured from the peak of the after-hyperpolarization to the AP peak, Max. depol. rate: maximum depolarization rate derived from the AP phase plane plot. Max. repol. rate: maximum repolarization rate derived from the AP phase plane plot, AHP: peak of the AP after-hyperpolarization, $t_m$: membrane time constant, $R_{in}$: input resistance, $C_m$: membrane capacitance.

exhibited significantly higher firing rates than AId neurons at high temperatures (*Figure 3B and D*; *Table 1*). Moreover, we consistently observed lower firing rates in AId neurons (*Figure 3B and D*), with lower instantaneous (frequency of the first two spikes) and steady-state (frequency of the last two spikes) firing frequencies at all levels of current injected (*Figure 3A and C*, top of each AP train). These results suggest that compared to RAPNs, the slower repolarization rate of AId neurons is likely a limiting factor for high-frequency repetitive spiking.

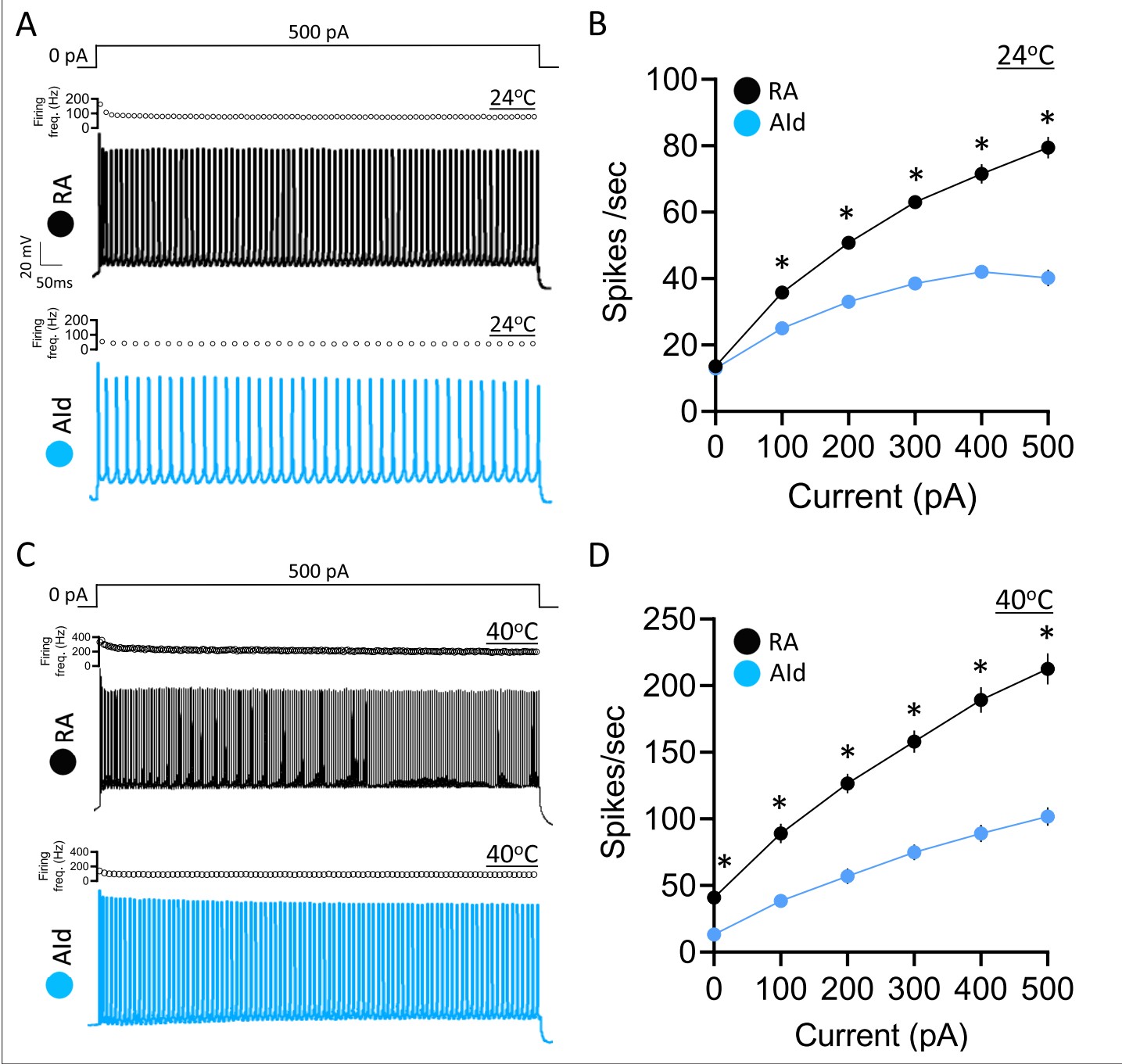

**Figure 3.** Evoked action potential (AP) properties in robustus arcopallialis projection neurons (RAPNs) and dorsal intermediate arcopallium (AId) neurons from adult male zebra finches at room and physiological temperatures. (**A**) Representative AP trains elicited by a 1 s 500 pA current injection at ~24°C in a RAPN (black, top) in an AId neuron (blue, bottom). The corresponding plot of firing frequency as function of time is shown at the top of each AP train. (**B**) Average elicited firing rate (spikes/s) as a function of injected current at ~24°C. Two-way ANOVA; $p=5.5 \times 10^{-22}$, $F_{(5,438)}=24.76$, N=39 RAPNs and 36 AId neurons. (**C**) Representative AP trains elicited by a 1 s 500 pA current injection at ~40°C in an RAPN (black, top) and in an AId neuron (blue, bottom). Same scale as in (**A**). The corresponding plot of firing frequency as function of time is shown at the top of each AP train. (**D**) Average elicited firing rate (spikes/s) as a function of injected current at ~40°C. Two-way ANOVA; $p=3.9 \times 10^{-7}$, $F_{(5,138)}=8.69$, N=13 RAPNs and 12 AId neurons. * in B & D indicates $p<0.05$; Tukey's post hoc test.

## Spikes are more sensitive to Kv3 blockers in RAPNs than in AId neurons

Kv3 channels (Kv3.1–3.4) are part of the Shaw-related family of voltage-gated K+ channels, which are characterized by rapid activation and deactivation kinetics at depolarized membrane potentials (*Kaczmarek and Zhang, 2017*; *Rudy and McBain, 2001*). These properties enable the rapid repolarization of APs, which in turn facilitates voltage-gated Na+ (Nav) channel recovery from inactivation during repetitive firing (*Bean, 2007*; *Leão et al., 2005*). Kv3.1, in particular, has been implicated in narrowing the AP waveform in a number of cell types, including Betz cells (*Ichinohe et al., 2004*; *Soares et al., 2017*). Interestingly, *KCNC1* (Kv3.1) mRNA is expressed within RA (*Lovell et al., 2013*). Moreover, RA shows much higher expression of *KCNC1* than AId (*Nevue et al., 2020*). We thus asked whether Kv3.1 channels are regulators of the AP half-width in RAPNs and AId neurons. Although there are no commercially available Kv3.1-specific inhibitors, previous studies have established pharmacological protocols to confirm the presence of Kv3.x currents in excitable cells (*Liu et al., 2017*; *Muqeem et al., 2018*). APs from Kv3.x-expressing neurons, including Betz cells (*Spain et al., 1991*), are typically broadened by sub-millimolar concentrations of the Kv channel inhibitors tetraethylammonium (TEA) and 4-aminopyridine (4-AP) (*Rettig et al., 1992*). If the differences in AP half-width and

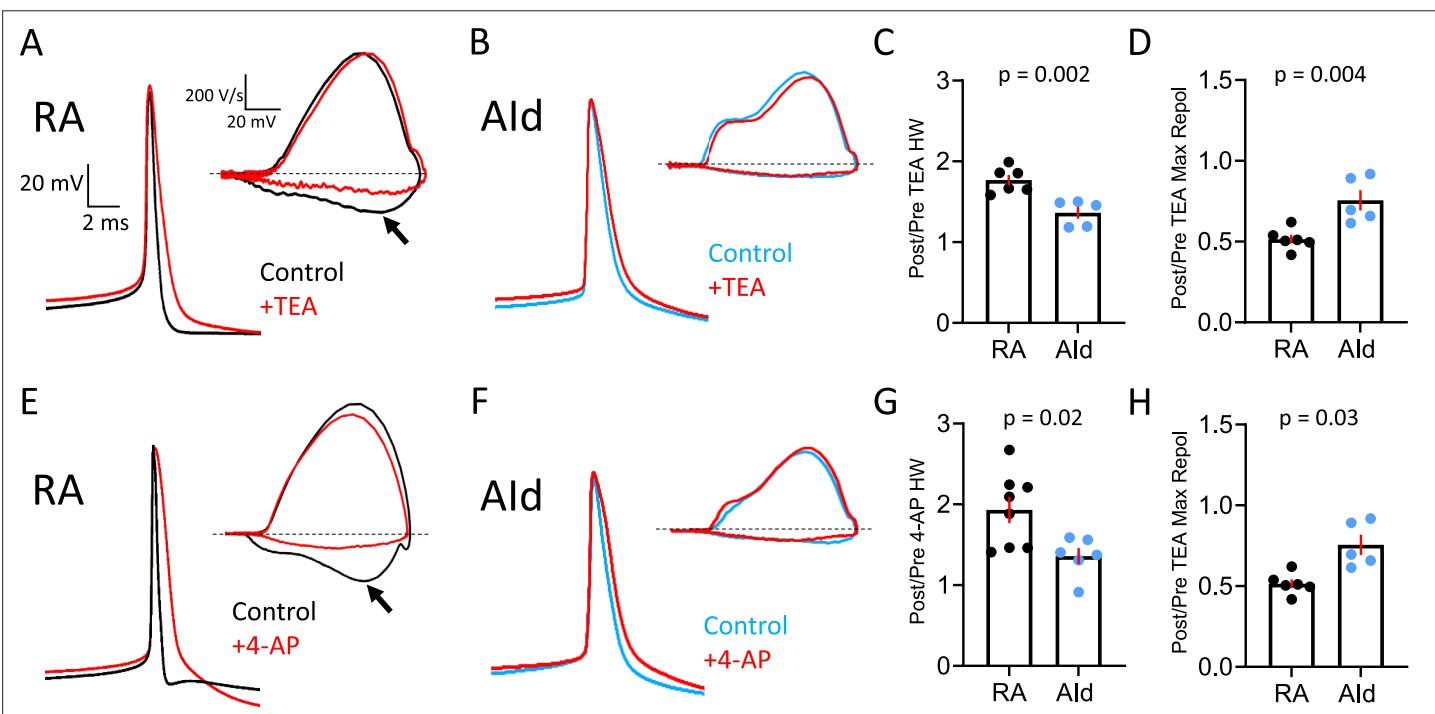

**Figure 4.** Effect of Kv3 channel inhibitors on spontaneous action potentials (APs) of robustus arcopallialis projection neuron (RAPN) and dorsal intermediate arcopallium (AId) neurons. (**A**) Representative averaged spontaneous RAPN AP traces and phase plane plots (inset) before and after 500 µM tetraethylammonium (TEA) administration. (**B**) Same as in (**A**) with AId neurons. (**C**) Fold change in the AP half-width (HW) in RAPNs and AId neurons (or ratio of post/pre TEA treatment). Student's t-test, two-tailed, $t_{stat}$ = 4.184, N=6 RAPNs and 5 AId neurons. (**D**) Fold change in the AP maximum repolarization rate in RAPNs and AId neurons. Student's t-test, two-tailed, $t_{stat}$ = 3.782, N=6 RAPNs and 5 AId neurons. (**E**) Representative averaged spontaneous RAPN AP traces and phase plane plots (inset) before and after 100 µM 4-aminopyridine (4-AP) administration. (**F**) Same as in (**A**) with AId neurons. (**G**) Fold change in the AP half-width in RAPNs and AId neurons. Student's t-test, two-tailed, $t_{stat}$ = 2.759, N=7 RAPNs and 6 AId neurons. (**H**) Fold change in the AP maximum repolarization rate in RAPNs and AId neurons. Student's t-test, two-tailed, $t_{stat}$ = 2.499, N=7 RAPNs and 6 AId neurons. p-Values are included in the graphs, red bars indicate standard error in C–D and G–H. The black arrows in A and E point to the changes in the maximum rate of repolarization. Dashed lines in A–B and E–F represent 0 V/s.

The online version of this article includes the following figure supplement(s) for figure 4:

**Figure supplement 1.** Kv3.1 antagonists broaden the robustus arcopallialis projection neuron (RAPN) and dorsal intermediate arcopallium (AId) neuron spontaneous action potentials (APs).

**Figure supplement 2.** Inhibitors of distinct tetraethylammonium (TEA)-sensitive K+ channels do not differentially affect robustus arcopallialis projection neuron (RAPN) and dorsal intermediate arcopallium (AId) neuron spontaneous action potentials (APs).

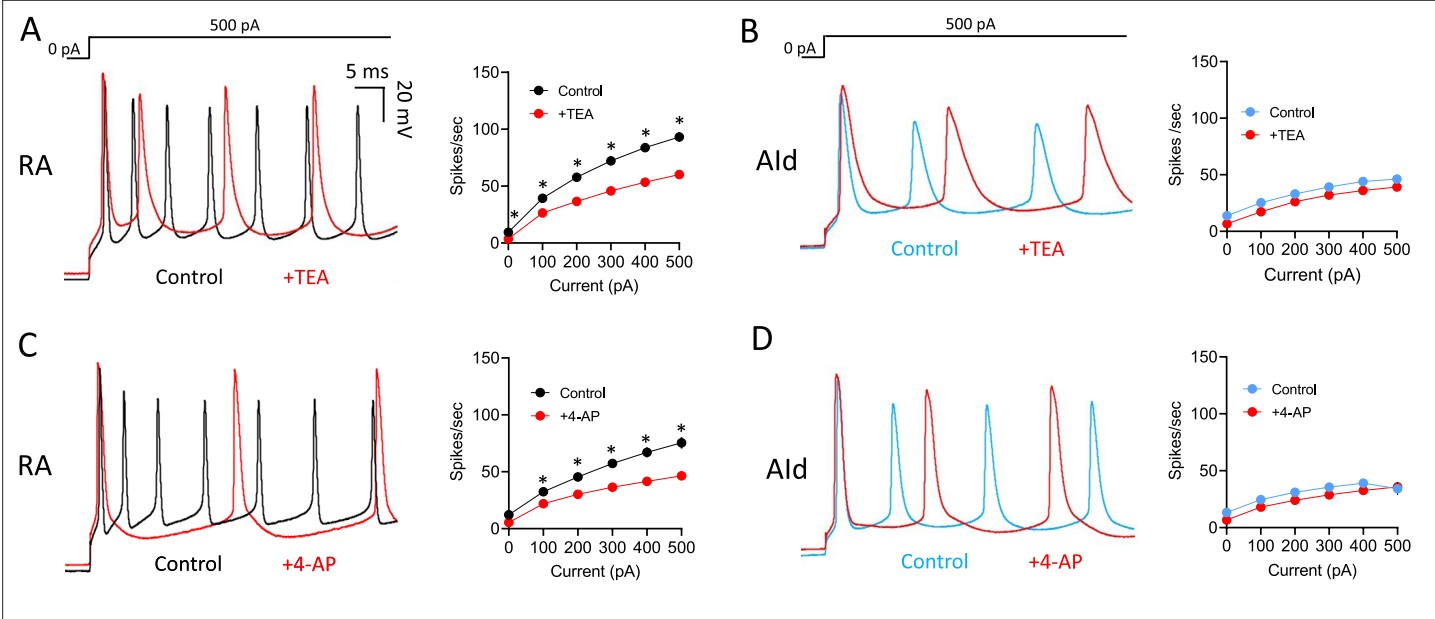

**Figure 5.** Effects of Kv3 channel inhibitors on evoked action potentials (APs) in robustus arcopallialis projection neurons (RAPNs) and dorsal intermediate arcopallium (AId) neurons. (**A**) Left to right: Representative first 50 ms of AP traces elicited by a 1 s 500 pA current injection and average firing rates (spikes/s) before and after exposure of RAPNs to 500 μM tetraethylammonium (TEA). Repeated measures two-way ANOVA; p=6.0 × 10⁻⁴, F(1.279,7.672)=27.69, N=7 RAPNs. (**B**) Same as in (**A**) except in AId neurons. Repeated measures two-way ANOVA; p=0.73, F(1.863,11.18)=0.3056, N=7 AId neurons. (**C**) Left to right: Representative first 50 ms of AP traces elicited by a 1 s 500 pA current injection and average spikes/s before and after exposure of RAPNs to 100 μM 4-aminopyridine (4-AP). Repeated measures two-way ANOVA; p=8.0 × 10⁻⁵, F(1.256,8.794)=41.87, N=8 RAPNs. (**D**) Same as in (**C**) except in AId neurons. Repeated measures two-way ANOVA; p=0.26, F(1.133,5.665)=1.575, N=6 AId neurons. Scales are the same in A–D. * in A–D indicates p<0.05; Tukey's post hoc tests.

The online version of this article includes the following figure supplement(s) for figure 5:

**Figure supplement 1.** Washout of tetraethylammonium (TEA) and 4-aminopyridine (4-AP) from recordings from robustus arcopallialis projection neurons (RAPNs).

**Figure supplement 2.** Inhibitors of tetraethylammonium (TEA)-sensitive channels do not differentially affect robustus arcopallialis projection neuron (RAPN) and dorsal intermediate arcopallium (AId) neuron evoked action potentials (APs).

maximum repolarization rate between RAPNs and AId neurons are due to the expression of Kv3.1, we would expect a more significant AP broadening upon exposure to either of these compounds in RAPNs than in AId neurons.

To test this prediction, we recorded spontaneous APs from RAPNs and AId neurons in frontal slices before and after exposure to 500 μM TEA (**Figure 4A–B**) and 100 μM 4-AP (**Figure 4E–F**), respectively. We performed these experiments at room temperature, as this allowed for more stable recordings that lasted for longer time periods. In response to TEA, spontaneous APs in RAPNs showed significantly more broadening (**Figure 4C**; mean ± SEM [ms]: 0.53±0.01–0.92±0.04 [77% change] and 1.24±0.06–1.71±0.15 [36% change] for RAPNs and AId neurons, respectively; for individual measurements in RAPNs and AId neurons, see **Figure 4—figure supplement 1A–B**, left) and decreases in the maximum repolarization rate (**Figure 4D**; mean ± SEM [ms]: 196.8±7.7–101.8±8.0 [49% change] and 51.6±2.5–39.5±5.1 [24% change] for RAPNs and AId neurons, respectively; for separate measurements in RAPNs and AId neurons, see **Figure 4—figure supplement 1A–B**, right). In response to 4-AP, spontaneous APs in RAPNs also showed significantly more broadening (**Figure 4G**; mean ± SEM [ms]: 0.53±0.04–0.98±0.04 [93% change] and 1.10±0.05–1.49±0.13 [36% change] for RAPNs and AId neurons, respectively; for individual measurements in RAPNs and AId neurons, see **Figure 4—figure supplement 1C–D**, left) and decreases in the maximum repolarization rate (**Figure 4H**; mean ± SEM [ms]: 185.9±21.7–109.9±3.4 [35% change] and 66.3±3.3–61.9±7.0 [8% change] for RAPNs and AId neurons, respectively; for individual measurements in RAPNs and AId neurons, see **Figure 4—figure supplement 1C–D**, right).

Evoked AP firing was also more affected by TEA and 4-AP in RAPNs than in AId neurons. Upon exposure to TEA, RAPNs showed significant decreases in the evoked firing rate (*Figure 5A*; 35 ± 4.1% decrease in spikes/s, 29.3 ± 10.4% decrease in instantaneous firing, and 33.7 ± 3.8% decrease in steady-state firing frequency upon the +500 pA current injection), whereas trends, but no significant effects, were seen in AId (*Figure 5B*). Upon exposure to 4-AP, RAPNs also showed significant decreases in the evoked firing rate (*Figure 5C*; 36.8 ± 10.9% decrease in spikes/s, 63.7 ± 4.8% decrease in instantaneous firing frequency, and 36.5 ± 12.9% decrease in steady-state firing frequency upon the +500 pA current injection), whereas trends, but no significant effects, were seen in AId (*Figure 5D*).

In order to confirm that the results obtained with TEA and 4-AP were not an artifact of time spent in the whole-cell current-clamp configuration, we performed washout experiments (*Figure 5—figure supplement 1*). We observed a washout of the effects of TEA within 3 min, both on the AP half-width (*Figure 5—figure supplement 1A–B*; 82% recovery) and the spikes per second during a +300 pA current injection (*Figure 5—figure supplement 1C*; 98% recovery). We additionally observed a trend for recovery from the effects of 4-AP on the AP half-width (*Figure 5—figure supplement 1D–E*) and spikes per second (*Figure 5—figure supplement 1F*) on a longer timescale than TEA. This was not surprising as 4-AP displays prolonged residency within cells by crossing of the plasma membrane (*Molgó et al., 1980*). These results confirm the stability of our recordings and the reversibility of TEA and 4-AP effects in our experiments.

Our results with TEA and 4-AP alone, however, could not rule out the possible contributions of other $K^+$ channels that are also sensitive to either TEA (Kv1, Kv7, and large-conductance $Ca^{2+}$-activated $K^+$ [BK] channels; *Al Sabi et al., 2010*; *Shen et al., 1994*; *Wang et al., 1998*) or 4-AP (Kv1; *Shu et al., 2007*; *Storm, 1988*; *Wu and Barish, 1992*). Thus, we next tested the effects α-dendrotoxin (DTX), XE991, and iberiotoxin (IbTX), which are highly selective antagonists of Kv1.1/1.2/1.6, Kv7, and BK channels, respectively. Upon exposing RAPNs and AId neurons to these antagonists, we observed no significant effects on spontaneous AP half-width or maximum repolarization rate (*Figure 4—figure supplement 2A–F*) in contrast to the effects of TEA and 4-AP. We also observed little to no effects on spontaneous and evoked firing frequency in RAPNs or AId neurons (*Figure 5—figure supplement 2A–F*). Importantly, we anticipate these compounds would produce significant effects if the indicated channels were expressed as we note high homology between the mammalian and avian amino acid sequences for these channels (sequence conservation ranged from 77% to 98% with ≥97% conservation between the first and last transmembrane segments for all channels), with previous studies demonstrating effects of both DTX (*Rathouz and Trussell, 1998*) and IbTX (*Moonen et al., 2010*) in avian species. Thus, we conclude that Kv3.x subunits are the predominant TEA- and 4-AP-sensitive channels in the recorded arcopallial cells.

## RAPNs have a faster, larger, high threshold TEA-sensitive $I_{K+}$ than AId neurons

To confirm that the regional differences in our current-clamp recordings were indeed due to differences in the outward voltage-gated $K^+$ current ($I_{K+}$), we performed voltage-clamp recordings. We note that our recordings are performed in frontal slices that transect RAPN axons near the soma. To further minimize space and voltage-clamp issues, we (1) decreased the slice thickness from 180 μm to 150 μm to eliminate more neuronal processes, (2) lowered intracellular $K^+$ from 142.5 mM to 75 mM to decrease the magnitude of $I_K$, and (3) compensated the series resistance electronically to 1 MΩ. We have previously shown that while these conditions do not offer complete space and voltage-clamp control, they significantly improve these parameters (see *Zemel et al., 2021*). After pharmacologically isolating $I_{K+}$, we delivered sequential 200 ms test pulses to –30 mV and 0 mV, from a 5 s holding potential at –80 mV, to preferentially activate low threshold $I_{K+}$ or both low and high threshold $I_{K+}$ respectively. Both RAPNs and AId neurons produced outward currents during both depolarizing steps (*Figure 6A*). Consistent with a difference in expression of a high threshold, non-inactivating $I_{K+}$ RAPNs produced significantly larger peak outward currents 200 ms ($I_{200ms}$) after stepping to 0 mV (*Figure 6A–B*; mean ± SE: 12.2±1.2 nA vs. 4.8±0.7 nA for RAPNs and AId neurons, respectively), whereas outward currents were indistinguishable between RA and AId during the –30 mV step (*Figure 6B*; mean ± SE: 0.8±0.1 nA vs. 0.4±0.04 nA for RAPNs and AId neurons, respectively). We also noted an initial A-type current waveform component in both RAPNs and AId neurons that preceded the larger delayed rectifier component during the 0 mV step (*Figure 6A*, red shaded area of the current at the 0 mV test

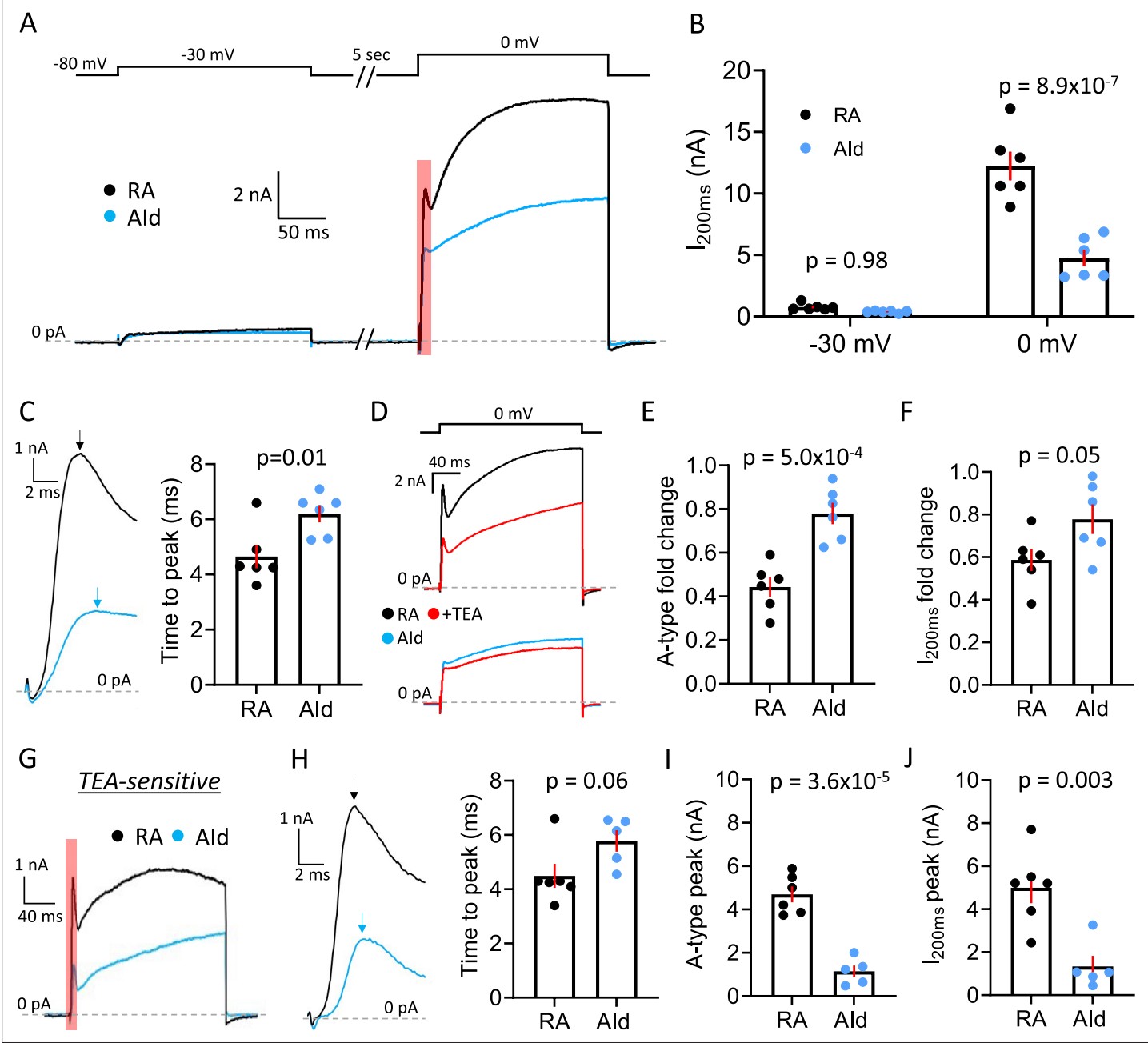

**Figure 6.** Comparison of $I_{K+}$ between robustus arcopallialis projection neurons (RAPNs) and dorsal intermediate arcopallium (AId) neurons. (**A**) Representative voltage-clamp recordings of $I_{K+}$ in an RAPN and an AId neuron at –30 mV and 0 mV. Shown at the top are the test pulses separated by 5 s at –80 mV, with a 5 s inter-sweep interval at –80 mV. (**B**) Comparison of the peak current at 200 ms ($I_{200ms}$) during both the –30 mV and 0 mV test pulses in RAPNs and AId neurons (two-way ANOVA: p=3.8 × 10$^{-5}$, F(1,20)=27.62, N=6 RAPNs and 6 AId neurons). Red bars indicate SEM. Individual p-values determined by Tukey's post hoc test. (**C**) Left: Expanded view of the A-type current component highlighted in (**A**). Arrows point to the peak of the A-type component of the current. Right: Comparison of the time to peak of both currents as measured from the onset of the voltage step to the peak of the A-type current (Student's t-test, $t_{stat}$ = 2.951, N=6 RAPNs and 6 AId neurons). (**D**) Representative voltage-clamp recordings of a RAPN (black; top) and AId neurons (blue; bottom) before and during exposure to 500 μM TEA (red) at the 0 mV test pulse. (**E**) Comparison of the fold change in the initial A-type peak of $I_{K+}$ elicited at 0 mV between RAPNs and AId neurons (Student's t-test, two-tailed, $t_{stat}$ = 5.062, N=6 RAPNs and 6 AId neurons). Red bars indicate SEM. (**F**) Comparison of the fold change in the $I_{200ms}$ of the $I_{K+}$ elicited at 0 mV between RAPNs and AId neurons (Student's t-test, two-tailed, $t_{stat}$ = 2.200, N=6 RAPNs and 6 AId neurons). Red bars indicate SEM. (**G**) Representative TEA-sensitive currents from the 0 mV test pulse in an RAPN (black) and an AId neuron (blue). (**H**) Left: Expanded view of the A-type current component highlighted in (**A**). Arrows pointing to the peak of the A-type component of the current highlighted in (**G**). Right: Comparison of the time to peak of both currents as measured from the onset of the voltage step to the peak of the A-type current (Student's t-test, $t_{stat}$ = 2.118, N=6 RAPNs and 5 AId neurons). (**I**) Comparison of the peak of the A-type component of the TEA-sensitive

*Figure 6 continued on next page*

*Figure 6 continued*

current (Student's t-test, $t_{stat}$ = 7.528, N=6 RAPNs and 5 AId neurons). Red bars indicate SEM. (**J**) Comparison of the $I_{200ms}$ peak (Student's t-test, $t_{stat}$ = 4.028, N=6 RAPNs and 5 AId neurons). Red bars indicate SEM.

The online version of this article includes the following figure supplement(s) for figure 6:

**Figure supplement 1.** Inactivation of tetraethylammonium (TEA)-sensitive $I_{K+}$ in robustus arcopallialis projection neurons (RAPNs) and dorsal intermediate arcopallium (AId) neurons.

pulse). Interestingly, the time to this peak was shorter in RAPNs than in AId neurons (*Figure 6C*; mean ± SE: 4.7±0.4 ms vs. 6.2±0.3 ms for RAPNs and AId neurons, respectively), even though the $I_{K+}$ size was much larger in the RAPNs.

Informed by our current-clamp pharmacology experiments, we next tested the prediction that $I_{K+}$ would be more attenuated by sub-millimolar concentrations of TEA in RAPNs than in AId neurons. Upon washing in 500 µM TEA onto slices we saw decreases in the peak current from neurons in both brain regions (*Figure 6D*), with RAPNs showing a larger fold change in both the peak of the A-type current component (*Figure 6E*; mean ± SE: 55.7±4.4% vs 22.0 ± 4.9% decrease for RAPNs and AId neurons, respectively) and in the peak current at $I_{200ms}$ (*Figure 6F*; mean ± SE: 41.2±5.2% vs 22.2 ± 7.0% decrease for RAPNs and AId neurons, respectively). We then extracted the TEA-sensitive $I_{K+}$ by subtracting the post-TEA traces from the pre-TEA traces. Like the pre-TEA currents obtained at 0 mV, the TEA-sensitive current in both RAPNs and AId neurons had A-type and delayed rectifier components (*Figure 6G*). By dividing the current 8 ms after the initial A-type peak by the peak A-type current, we found that the degree of inactivation was indistinguishable between RAPNs and AId neurons (*Figure 6—figure supplement 1*). Notably, the time to peak of the A-type component was preserved (mean ± SE: 4.5±0.4 ms vs. mean ± SE: 5.8±0.4 ms for RAPNs and AId neurons, respectively), with RAPNs maintaining a trend toward smaller values (*Figure 6H*). Importantly, whether measuring the A-type current peak (*Figure 6I*; mean ± SE: 4.7±0.4 nA vs. 1.1±0.3 nA for RAPNs and AId neurons, respectively) or the $I_{200ms}$ peak (*Figure 6J*; mean ± SE: 5.0±0.7 nA vs. 1.3±0.5 nA for RAPNs and AId neurons, respectively), RAPNs had a significantly larger TEA-sensitive $I_{K+}$ than AId neurons. In sum, our voltage-clamp results show a higher proportion of a fast-activating, TEA-sensitive current in RAPNs compared to AId neurons, consistent with the presence of a larger Kv3 current in RAPNs.

## Differential mRNA expression of the TEA-sensitive ion channel subunit Kv3.1 in RAPNs vs. AId neurons

The evidence presented thus far is consistent with the ultra-fast APs unique to RAPNs being related to higher expression of Kv3.1 in RA compared to AId. However, Kv3.1 is only one of four members of the Shaw-related channel family (Kv3.1–3.4) in vertebrates (*Kaczmarek and Zhang, 2017*; *Rudy and McBain, 2001*). To further examine a link between RAPN properties and Kv3.1, it was important to examine other Kv3.x family members. As a start, through close assessments of reciprocal alignments and synteny we confirmed that the locus named *KCNC1* (100144433; located on chromosome 5) is the zebra finch ortholog of mammalian *KCNC1*, noting the conserved synteny across major vertebrate groups (*Figure 7—figure supplement 1A*). Importantly, the predicted zebra finch Kv3.1 protein (Kv3.1b isoform) is remarkably conserved (96.5% residue identity) with human (*Figure 7—figure supplement 2*) and is thus predicted to have similar pharmacology as in mammals, which is supported by our recordings from RAPNs. Additionally, we confirmed the correct identification of the zebra finch orthologs of mammalian *KCNC2*/Kv3.2 and *KCNC4*/Kv3.4, as previously reported (*Lovell et al., 2013*).

Previous investigations of the zebra finch genome (taeGut1, *Warren et al., 2010*) reported that zebra finches lacked a *KCNC3*/Kv3.3 ortholog (*Lovell et al., 2013*). Recent long-read sequencing technology, however, has facilitated a more complete assembly of genomes (*Rhie et al., 2021*), elucidating the presence of some genes previously thought to be absent. Using RefSeq release 106 (GCF_003957565.2 assembly), we observed a locus (LOC115491734) described as similar to member 1 of the *KCNC* family on the newly assembled zebra finch chromosome 37, one of the smallest and hardest to sequence avian microchromosomes. LOC115491734, however, exhibited the highest alignment scores and conserved upstream synteny with *KCNC3*/Kv3.3 in humans and various vertebrate lineages, noting that NAPSA in zebra finch and other songbirds is misannotated as cathepsin D-like

(LOC121468878) (*Figure 7—figure supplement 1C*). We conclude that LOC115491734 is the zebra finch ortholog of mammalian *KCNC3*, previously thought to be missing in birds (*Lovell et al., 2013*), and not *KCNC1*. We have also identified an avian *KCNC3* locus in a few other songbird and non-songbird species, noting that in most cases the gene is incorrectly annotated in NCBI as *KCNC1* or *KCNC1*-like, whereas conversely, a large set of avian genes in this family are incorrectly annotated as *KCNC3*-like (*Figure 7—figure supplement 1B–C*). Also notably, a *KCNC3* locus (XM_009582050.1) previously described as the ortholog of *KCNC3* in two seabirds (*De Paoli-Iseppi et al., 2017*) was misidentified in that study and most likely represents *KCNC4* in those species. Interestingly, the downstream immediate synteny is not conserved across vertebrate lineages, with the ancestral condition in tetrapods likely being TBC1D17 downstream of *KCNC3*. The predicted zebra finch *KCNC3* protein showed only moderate conservation with human (68.32% residue identity), some domains including the BTB/POZ and transmembrane domain being fairly conserved, but spans of residues on the N-terminal and C-terminal regions being highly divergent. Notably, based on the genomic sequence, the N-terminal inactivation domain (*Rudy and McBain, 2001*) is absent in the predicted zebra finch *KCNC3* protein (*Figure 7—figure supplement 3*).

We next performed in situ hybridization for all identified *KCNC*/Kv3.x family members in adjacent frontal brain sections from adult male zebra finches. We replicated our previous finding that *KCNC1*/Kv3.1 expression is higher in RA than in AId (*Nevue et al., 2020*; *Figure 7A*, top left), whereas expression of both Kv3.2 (*Figure 7A*, top right) and Kv3.4 (*Figure 7A*, bottom right) was non-differential between RA and AId. Unlike the graded Kv3.1 expression that matches a tonotopic distribution found in avian (*Parameshwaran et al., 2001*) and mammalian (*Li et al., 2001*) auditory brainstem, Kv3.1 expression was uniformly distributed across RA. Kv3.2-expressing cells were sparse, strongly labeled, and reminiscent of the GABAergic cell distribution (*Pinaud and Mello, 2007*), while Kv3.4 expression was uniformly weak throughout both brain regions (*Figure 7A*). We also found that Kv3.3, while more strongly expressed in both brain regions compared to the surrounding arcopallium, is non-differentially expressed between RA and AId (*Figure 7A*, bottom left). Importantly, we found strong Kv3.3 expression in the Purkinje cell layer of the cerebellum (*Figure 7A*, bottom left), consistent with findings in mammals (*Akemann and Knöpfel, 2006*). Furthermore, other TEA-sensitive potassium channel subunits examined, namely members of the Kv1 (*KCNA*), BK (*KCNMA*), and Kv7 (*KCNQ*) families, had similar expression in RA and AId, with the exception of *KCNQ2*, which had a lower proportion of labeled cells in RA than in AId (*Figure 7B and C*). These results indicate that compared to the other TEA-sensitive channels, *KCNC1*/Kv3.1 is the only TEA-sensitive subunit we examined that was more highly expressed in RA compared to AId, providing supporting evidence for a stronger contribution of Kv3.1 in shaping the ultranarrow AP of RAPNs compared to AId neurons.

Intriguingly, while curating avian *KCNC3s*, we discovered a previously undescribed *KCNC* family member in the genomes of several bird species, but notably absent in songbirds (*Figure 7—figure supplement 1B*), a finding that cannot be explained by gaps in genomic sequence in the latter. This gene most closely resembled *KCNC1* in predicted domains and amino acid conservation, thus we named this *KCNC1* paralog as *KCNC1*-like (*KCNC1L*). We also observed *KCNC1L* in non-avian sauropsids including lizards and snakes, where the locus seems to be duplicated. It was not present in humans/mammals, nor in amphibian or fish outgroups. This suggests this paralog possibly arose after the split between mammals and sauropsids, with a subsequent loss in songbirds. Overall, while our comparative analysis helps to further link the differential expression of *KCNC1*/Kv3.1 to physiological differences between RA and AId, it also brings new insights into the evolution of this key family of neuronal excitability regulators. How the newly identified avian *KCNC3* and non-oscine *KCNC1L* contribute to avian neuronal physiology are intriguing questions for future studies.

## AUT5 narrows the AP half-width and increases the firing rate of RAPNs

Whereas pharmacological inhibitors provided strong evidence of a major role for Kv3.x channels in RAPNs compared to AId neurons (*Figures 4–6*), they notably lack the specificity for Kv3.x subunits. In contrast, the specific (*Covarrubias et al., 2023*), novel positive Kv3.1/3.2 modulator, AUT5, potentiates Kv3.1 currents by speeding up activation kinetics (*Taskin et al., 2015*) and leftward shifting the voltage dependence of activation (*Taskin et al., 2015*; *Covarrubias et al., 2023*). To examine AUT5 effects, we recorded both spontaneous (*Figure 8A and C*, left) and evoked (*Figure 8B and D*, left) APs from RAPNs and AId neurons in the whole-cell current-clamp configuration, before

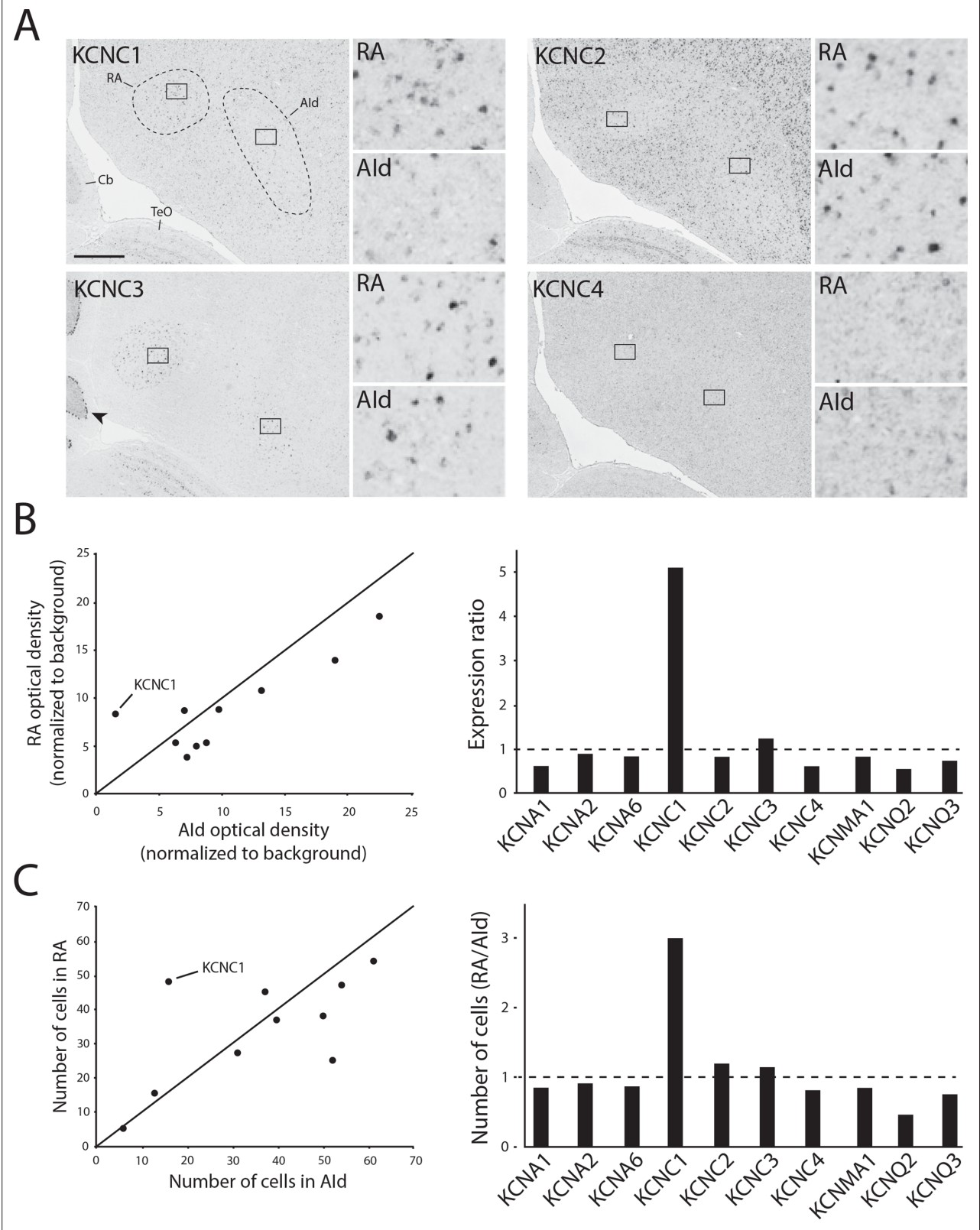

**Figure 7.** K+ channel diversity in the zebra finch arcopallium: Stronger expression of *KCNC1* (Kv3.1 subunit) transcripts in RA than in dorsal intermediate arcopallium (AId). (**A**) Representative in situ hybridization images for Kv3 channel family member transcripts in RA (left) and AId (right), from nearly adjacent frontal sections of adult males. Squares in large images depict position of counting windows and of inset images for RA and AId. Arrow points to strong Kv3.3 mRNA staining in the Purkinje cell layer in the cerebellum. *Cb* - cerebellum, *TeO* - optic tectum. Scale bar: 500 µm. (**B**) Optical density

*Figure 7 continued on next page*

*Figure 7 continued*

measurements (background subtracted) in RA and AId for subunits associated with tetraethylammonium (TEA)-sensitive Kv channel types. Expression ratio (right) was calculated as $RA_{OD}/AId_{OD}$. (**C**) Labeled cell counts in RA and AId for subunits associated with TEA-sensitive Kv channel types. Cell count ratios of RA/AId are shown on right.

The online version of this article includes the following figure supplement(s) for figure 7:

**Figure supplement 1.** Comparative genomics of *KCNC*/Kv3 family members.

**Figure supplement 2.** Amino acid alignment between Kv3.1 in human (top) and zebra finch (bottom).

**Figure supplement 3.** Amino acid alignment between Kv3.3 in human (top) and zebra finch (bottom).

and during exposure to 1 µM AUT5 ($EC_{50}$ = 1.3 µM, *Taskin et al., 2015*). Compared to pre-drug measurements, RAPNs showed a significant narrowing of the AP waveform (mean ± SE: 6.5 ± 2.1% decrease; *Figure 8A*, middle) and an increase of the maximum repolarization rate (mean ± SE: 12.2 ± 4.3% increase; *Figure 8A*, right) but not of the maximum depolarization rate (paired t-test, $t_{stat}$ = 1.04, p=0.3), whereas no significant effects were seen in AId neurons (*Figure 8C*, middle and right). Spontaneous APs from RAPNs also displayed significant depolarizations in the peak (mean ± SE: 35.1±1.1 mV to 38.9±1.2 mV, paired t-test, $t_{stat}$ = 2.72, p=0.03), threshold (mean ± SE: −58.9±1.8 mV to −57.0±1.6 mV, paired t-test, $t_{stat}$ = 3.542, p=0.01), with a non-significant trend observed for the peak after-hyperpolarization (mean ± SE: −72.0±0.7 mV to −71.1±0.7 mV, paired t-test, $t_{stat}$ = 2.3, p=0.06). We note, that the depolarization in the AP peak could in turn further recruit Kv channel-mediated AP repolarization. In comparison AId only showed a modest change in the after-hyperpolarization (−69.6±1.1 to −67.3±0.6, paired t-test, $t_{stat}$ = 2.874, p=0.03). These changes in the spike waveform correlated with a significant increase in evoked spikes produced in RAPNs that was not seen in AId neurons (*Figure 8B and D*; mean ± SE: 25.7% ± 6.1% for RAPNs during the 1 s + 500 pA current injection). This effect is consistent with AUT5 effects observed in the Kv3.1 expressing hippocampal

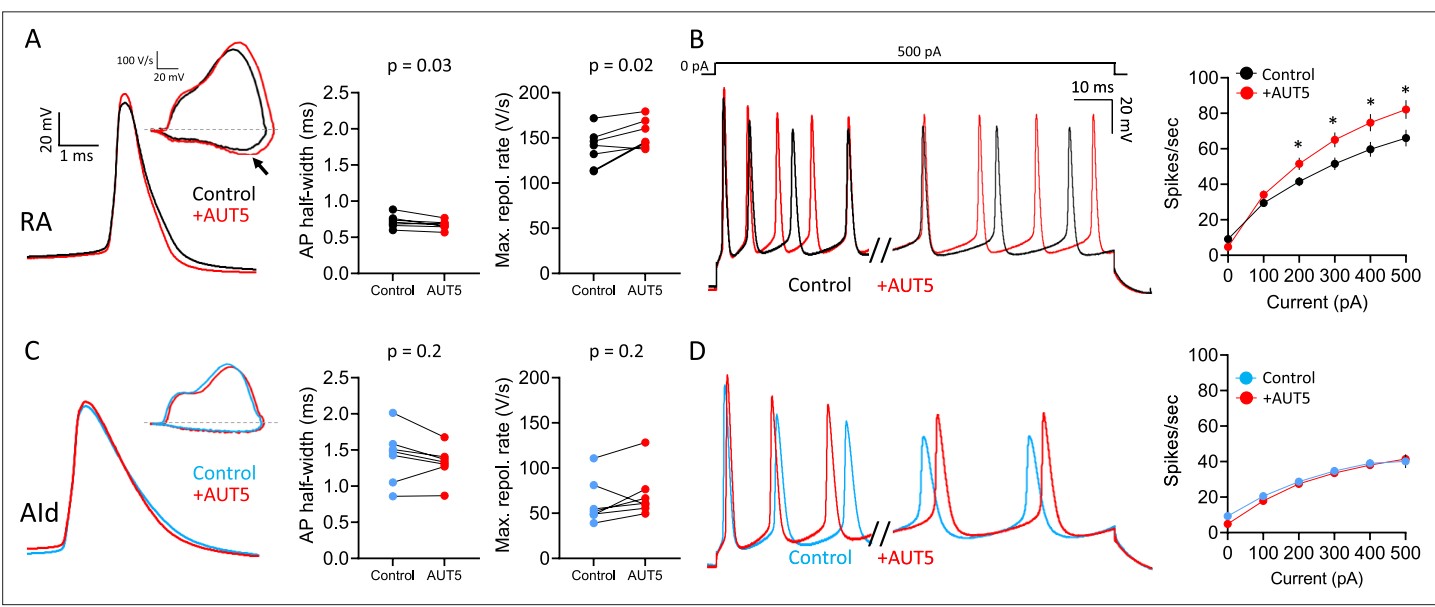

**Figure 8.** Effects of the Kv3.1/3.2 positive modulator AUT5 on action potentials (APs) of robustus arcopallialis projection neuron (RAPN) and dorsal intermediate arcopallium (AId) neurons. (**A**) Left to right: Representative AP traces, phase plane plots (inset), and changes in AP half-width and maximum repolarization rate upon exposure to 1 µM AUT5 for RAPNs. The black arrow points to the change in the maximum rate of repolarization. AP half-width: paired Student's t-test, two-tailed, $t_{stat}$ = 2.838, N=7 RAPNs; maximum repolarization rate: paired Student's t-test, two-tailed, $t_{stat}$ = 3.095, N=7 RAPNs. (**B**) Left to right: First 40 ms and last 65 ms of representative AP traces elicited by a 1 s 500 pA current injection and average evoked firing rates (spikes/s) before and after exposure to 1 µM AUT5 for RAPNs. Repeated measures two-way ANOVA; p=3.1 × $10^{-12}$, F(5,30)=32.40, N=7 RAPNs; * indicates p<0.05; Tukey's post hoc tests. (**C**) Left to right: Representative AP traces, phase plane plots (inset), and changes in AP half-width and maximum repolarization rate upon exposure to 1 µM AUT5 for AId neurons. AP half-width: paired Student's t-test, $t_{stat}$ = 1.483, N=7 AId neurons, two-tailed; maximum repolarization rate: paired Student's t-test, two-tailed, $t_{stat}$ = 1.523, N=7 AId neurons. Same scale as in (**A**). (**D**) Left to right: First 40 ms and last 65 ms of representative AP traces elicited by a 1 s 500 pA current injection and average evoked firing rates (spikes/s) before and after exposure to 1 µM AUT5 for AId neurons. Repeated measures two-way ANOVA; p=0.18, F(5,30)=1.658, N=7 AId neurons. Same scale as in (**B**).

GABAergic interneurons of rats (*Boddum et al., 2017*). Interestingly, whereas AUT5 had no effects on the instantaneous firing frequency in RAPNs, the steady-state firing frequency (as measured for the last two spikes recorded during the 1 s + 500 pA current injection) increased substantially (mean ± SE: 24.3 ± 8.2% increase). We did not observe significant washout with AUT5. This result was not surprising as previous studies indicate limited capacity for washout of this compound, even within cell lines (*Taskin et al., 2015*). Taken together, these results are again consistent with the finding of higher Kv3.1 expression in RAPNs than in AId neurons, and further support the role for Kv3.1 in the specialized fast firing properties of RAPNs compared to AId neurons.

## Discussion

Our results demonstrate that RAPNs and AId neurons share several properties, including spontaneous firing, minimally adapting firing during current injections and similar expression profiles of *KCNC2–4* (Kv3.2–3.4) subunits. However, they differ greatly in AP spike half-width and capacity for high-frequency firing. We show that these unique RAPN properties, which are reminiscent of the large Betz cells in layer 5 of primate M1 cortex, are associated with higher expression of *KCNC1* (Kv3.1) in RA than in AId. In contrast, the properties of AId neurons, which are involved in non-vocal somatic motor function in birds, are more similar to those of canonical mammalian L5PNs. We propose that RAPNs are highly specialized neurons for song production, sharing with primate Betz cells some unique physiological and molecular properties that allow both cell types to reliably fire ultranarrow spikes at high frequencies.

### Songbird upper motor neurons have distinct morphologies

The morphology of RAPNs described here is consistent with previous morphological characterizations, suggesting that these neurons represent a fairly homogeneous cell type (*Kittelberger and Mooney, 1999*; *Spiro et al., 1999*). In contrast, this study includes a first attempt to describe morphological features of AId neurons (*Figure 1*; *Figure 1—figure supplement 1*). We found that RAPNs and AId neurons share similar soma size and elaborate branched dendrites, and both form extensive local axon collaterals. However, we also obtained evidence of morphological differences that may have implications for how RAPNs and AId neurons integrate synaptic inputs. The higher dendritic branch complexity revealed by a Sholl analysis (*Sholl, 1956b*) and the higher number of dendritic spines (*Figure 1A*; *Figure 1—figure supplement 1C*) of AId neurons (a finding revealed using two methods: FIJI, *Schindelin et al., 2012*, and ShuTu, *Jin et al., 2019*) suggest that AId may receive more synaptic inputs than RAPNs. Such inputs would be from the nidopallium, the main known source of input to the AI (*Bottjer et al., 2000*; *Johnson et al., 1995*; *Karten, 2015*), and/or from local GABAergic interneurons. Conversely, the higher proportion of large spines in RAPNs than in AId neurons (*Figure 1—figure supplement 1D*) parallels Betz cells, which also contain higher proportions of large spines compared to other pyramidal neurons within the cat M1 (*Kaiserman-Abramof and Peters, 1972*). These differences in spine shape and surface area suggest potential differences in filtering of incoming synaptic signals, with some input sources potentially having larger effects on membrane potential changes than others. Of note, the spine density difference between RAPNs and AId neurons closely approximates findings in the high vocal center (HVC), where upstream HVC-RA projecting neurons exhibit roughly half the spine density as HVC-Area X projecting neurons (*Kornfeld et al., 2017*). Importantly, whereas axonal projection targets for RAPNs are discrete and have been studied in detail (*Vicario, 1991*; *Wild, 1993a*; *Wild et al., 2009*), those for AId neurons appear to be more complex (*Bottjer et al., 2000*) and are possibly more heterogeneous in terms of cell-type composition. While our present findings do not address potential subpopulations of AId neurons, they lay the groundwork for future efforts using cell filling and/or track tracing to further characterize the similarities and differences between RAPNs and AId neurons.

Calculations of membrane capacitance ($C_m$) allowed us to estimate the average surface area of RAPNs and AId neurons (*Table 1*), whereas 3D reconstructions using ShuTu allowed us to also estimate shrinkage-corrected surface areas (*Figure 1*, *Supplementary file 1*). The values were fairly close, in spite of the uncertainties and assumptions involved in both methods (see Results and Methods). Based on their larger overall spine surface area, it appears that AId neurons are investing more 'resources' on spines than RAPNs, despite having proportionately smaller individual spines. We suggest that the

sparse population of large spines in adult male RAPNs may indicate larger synaptic inputs per spine compared to AId neurons, a factor that could at least partly contribute to the highly stereotyped song of adult male finches. Large, stable and mature spines in RAPNs may thus enable large EPSCs in adult RAPNs from male birds with crystallized songs (*Garst-Orozco et al., 2014*; *Kittelberger and Mooney, 1999*). Future electrophysiology and morphology studies are needed to address this question further.

## Shared properties between upper motor neuron subclasses in finches and mammals

The speed and accuracy of fine muscle control in mammals requires the firing of L5PNs that project from the primary motor cortex (M1) to various targets in the brainstem and spinal cord (*Oswald et al., 2013*; *Suter et al., 2013*). We have recently described how upper motor RAPNs and L5PNs in mammalian M1 also share factors determining AP initiation and upstroke. This includes APs with biphasic depolarization rates, persistent Na$^+$ currents, large transient Na$^+$ currents ($I_{NaT}$) with rapid kinetics, and high Navβ4 mRNA expression, which we showed to be linked to robust resurgent currents ($I_{NaR}$) in RAPNs (*Zemel et al., 2021*). A major difference, however, was the ultranarrow AP waveform of RAPNs, which is also a unique trait of the large Betz cells compared to other L5PNs of M1 in cats (*Chen et al., 1996*) and primates (*Vigneswaran et al., 2011*). Here, we show that RAPNs produce narrower APs compared to AId neurons, largely due to higher maximum repolarization rates (*Figure 2*). In contrast to RAPNs, the 1.2 ms AP half-width at 24°C and the 'regular' minimally adapting AP firing of AId neurons render them more similar to canonical L5PNs in the motor cortex of rodents (*Lacey et al., 2014*), cats (*Chen et al., 1996*), and primates (*Vigneswaran et al., 2011*). Previous comparative studies in the auditory system have suggested convergent strategies for temporal coding of sound stimuli in birds and mammals (*Carr and Soares, 2002*; *Spool et al., 2021*). Our data suggest similar convergence in the motor control circuitry where, like Betz cells (*Bakken et al., 2021*; *Ichinohe et al., 2004*; *Soares et al., 2017*), RAPNs appear to be a specialized class of upper motor neurons that display higher temporal precision of AP firing and faster firing rates than AId neurons.

We also note, however, some marked differences between the properties of finch RAPNs and AId neurons compared to those of L5PNs in mammalian M1. Foremost, RAPNs and AId neurons lack the large, tufted apical dendrites typical of L5PNs (*Callaway et al., 2021*). Additionally, these avian neurons fire spontaneously in the absence of synaptic inputs, a property not typically seen in M1 L5PNs (*Chen et al., 1996*; *Lacey et al., 2014*). The size and distribution of Betz cells in M1 also differ compared to RAPNs in the finch. Whereas Betz cells have very large somas and are interspersed with smaller L5PNs (*Lassek, 1940*; *Rivara et al., 2003*), RAPNs and AId neurons have similar soma sizes and localize to adjacent but distinct regions within the finch arcopallium (*Bottjer et al., 2000*; *Johnson et al., 1995*; *Nevue et al., 2020*).

## Optimized for speed: Kv3.1 subunits facilitate fast spiking

By quickly activating at depolarized voltages, Kv3 channels can efficiently initiate Nav channel recovery from inactivation during repetitive AP firing (*Bean, 2007*; *Gu et al., 2018*; *Kaczmarek and Zhang, 2017*; *Rudy and McBain, 2001*). Importantly, the fact that the AP waveforms and firing rates of RAPNs were indeed more sensitive to sub-millimolar concentrations of TEA and 4-AP than AId neurons (*Figures 4–6*) implicates Kv3.1 channels in regulating the excitable features of RAPNs. Notably, however, these antagonists have known effects on other members of this ion channel family, even at sub-millimolar concentrations (*Kaczmarek and Zhang, 2017*). In situ hybridization showed higher expression of the Kv3.1 subunit in RAPNs than in AId neurons (*Figure 7*), suggesting a more specialized role of Kv3.1 in RAPNs compared to other upper motor neurons. Using the Kv3.1/3.2 positive modulator AUT5, we were able to further correlate the differential expression of Kv3.1 subunits with excitable properties in RAPNs and AId neurons. AUT5 is known to alter gating kinetics, and leftward shift the voltage dependence of activation of Kv3.1 channels (*Boddum et al., 2017*; *Covarrubias et al., 2023*; *Taskin et al., 2015*). The narrowing of the AP waveform and increase in steady-state firing provide support for a stronger contribution of Kv3.1 in the repolarization of APs and in increasing the availability of Nav channels during high-frequency firing in RAPNs versus AId neurons.

We also noted a modest depolarization of the AP peak, after-hyperpolarization and threshold with AUT5 exposure. Considering this combination of changes was not observed in AId, this may be an additional result of increases in Nav channel availability due to positive modulation of Kv3.1. The fact

that the AUT5 effects on Kv3.1-expressing zebra finch neurons were similar to those seen in mammals (*Boddum et al., 2017*) is not surprising, as the predicted peptide sequence of finch Kv3.1 is ~96.5% identical to that of the human Kv3.1b splice variant (*Figure 7—figure supplement 2*). Birds and mammals diverged 300 million years ago, so this remarkable conservation suggests that Kv3.1 channels may be optimized for enabling ultranarrow AP waveforms and high-frequency firing. Accordingly, loss-of-function mutations in the human Kv3.1 gene result in myoclonus epilepsy and ataxia, a disease that among other symptoms presents with severe motor deficits (*Barot et al., 2020*; *Muona et al., 2015*).

Kv3.1 channels have been found in fast-spiking, parvalbumin-expressing interneurons and in layer 5 Betz cells of primate motor cortex (*Bakken et al., 2021*; *Soares et al., 2017*). The two splice variants described in mammals, Kv3.1a and 3.1b, have differing trafficking patterns and protein-protein interactions (*Kaczmarek and Zhang, 2017*; *Rudy et al., 1999*). The longer C-terminal domain of Kv3.1b appears to enable trafficking out of the soma into the axon, while providing protein kinase C phosphorylation sites that decrease the open probability of the channel. We note that in previous transcriptome sequencing efforts in the finch, five Kv3.1 transcripts were predicted (NCBI GeneID:100144433), including Kv3.1b, that contain C-terminal domains with varying lengths. Thus, there is a strong likelihood that, like mammals, RAPNs express different Kv3.1 splice variants that are differentially trafficked and/or phosphorylated based on the specific characteristics of their C-terminal domains.

In contrast to Kv3.1, the expression of Kv3.2–3.4 is not different between RA and AId. Unlike Kv3.1 and Kv3.2, Kv3.4 exhibits rapid inactivation at depolarized voltages (*Kaczmarek and Zhang, 2017*; *Rudy and McBain, 2001*). Interestingly, whereas in mammals Kv3.3 channels exhibit inactivation, albeit on a slower timeframe than Kv3.4 (*Kaczmarek and Zhang, 2017*; *Rudy and McBain, 2001*), the predicted finch Kv3.3 peptide lacks an N-terminal 'ball-and-chain' domain thought to be associated with inactivation (*Figure 7—figure supplement 3*; *Rudy et al., 1999*). This would suggest that Kv3.4 may be solely responsible for the inactivating component of TEA-sensitive currents measured in both RA and AId (*Figure 6*; *Figure 6—figure supplement 1*), possibly by participating in a heteromultimeric complex with non-inactivating Kv3.1/2 subunits (*Baranauskas et al., 2003*). It is this initial part of the current, including the A-type component, that likely participates in the repolarization of APs in both RAPNs and AId neurons. Interestingly, cell-attached recordings in rat layer V pyramidal neurons in the sensorimotor cortex display TEA-insensitive A-type currents, a finding consistent with our data (*Kang et al., 2000*). This current is likely composed of channels that, in contrast to Kv3.4, gate at significantly more negative membrane potentials (*Kang et al., 2000*). Despite lacking an inactivation domain, Kv3.3 expressed in both RA and AId likely contributes to the TEA sensitivity seen in both current- and voltage-clamp recordings. The calyx of Held nerve terminal also generates ultranarrow spikes that can fire at 1 kHz (*Kim et al., 2013*) and both Kv3.1 and Kv3.3 subunits contribute to its excitability (*Richardson et al., 2022*). Importantly, these two subunits may be differentially trafficked to subcellular compartments in RA, including dendrites, which may facilitate burst-pause time coding (*Deng et al., 2005*; *Zang and De Schutter, 2021*).

## The combination of Kv3.1 and Navβ4 facilitates narrow and energetically efficient spikes

Birds and mammals are warm-blooded and their physiological temperatures facilitate narrow AP waveforms in a number of cell types by limiting the overlap of fast Na$^+$ and K$^+$ conductances, thus making the neurons more energetically efficient (*Alle et al., 2009*; *Fohlmeister, 2009*; *Hu et al., 2018*). RAPNs fire spontaneously with high-frequency bursts of spikes just before and during song production (*Daliparthi et al., 2019*; *Ölveczky et al., 2011*; *Yu and Margoliash, 1996*). This is presumably an energetically costly process and, accordingly, RA in adult males displays a dense staining for cytochrome oxidase (*Adret and Margoliash, 2002*).

The combination of Navβ4, and its associated $I_{NaR}$, and Kv3 channels likely promote ultranarrow AP waveforms and rapid bursting in RAPNs (*Zemel et al., 2021*) and fast-spiking nerve terminals (*Kim et al., 2010*; *Richardson et al., 2022*). Studies in rodent cerebellar Purkinje (*Akemann and Knöpfel, 2006*) and vestibular nucleus neurons (*Gittis et al., 2010*) suggest that the repolarization enabled by Kv3 currents enhances the activation of post-spike $I_{NaR}$, likely facilitating the high-frequency firing that occurs during in vivo bursting (*Loewenstein et al., 2005*; *Saito and Ozawa, 2007*). This is consistent for the recently identified role for $I_{NaR}$ in stabilizing burst duration and making neuronal firing

resistent to noise perturbations (*Venugopal et al., 2019*). This exquisite coordination between Navs, Kvs, and auxiliary subunits is likely part of a broader cohort of molecular markers identified as unique to fast-spiking neurons (*Hong and Sanchez, 2018*; *Kodama et al., 2020*; *Callaway et al., 2021*). Interestingly, arcopallial neurons outside of RA, and RAPNs from juvenile male finches have much lower expression of Navβ4 and Kv3.1 (*Nevue et al., 2020*; *Zemel et al., 2021*; *Friedrich et al., 2022*), exhibit broader APs, and are incapable of high-frequency firing (*Zemel et al., 2021*). Our results thus point to important synergistic roles of joint Navβ4 and Kv3.1 expression in shaping the remarkable RAPN excitable properties.

## Similarities between RAPNs and AId neurons and the evolutionary origins of RA

The AP waveforms in RAPNs and AId neurons are similar in several parameters, including threshold, maximum depolarization rate, amplitude, peak, and after-hyperpolarization (*Table 1*). These cells additionally have a multitude of common molecular correlates of excitability (e.g. high expression of Nav1.1, Nav1.6, and Navβ4; *Nevue et al., 2020*) that likely underlie some of their physiological similarities. We have now found that both RAPNs and AId neurons express Kv3.3 subunit transcripts, a previously unrecognized gene in the songbird genome, which likely imparts some of the TEA and 4-AP hypersensitivity to AP waveforms. These shared properties of RAPNs and AId neurons distinguish them from those examined in other arcopallial regions (e.g. caudal arcopallium outside of RA), where neurons exhibit broad APs, are incapable of sustained fast spiking, and exhibit less expression of the aforementioned ion channel subunits (*Friedrich et al., 2019*; *Nevue et al., 2020*; *Zemel et al., 2021*). Coupled with evidence of AId's involvement in somatic motor control (*Feenders et al., 2008*; *Yuan and Bottjer, 2020*), the current molecular and electrophysiological data further suggests that these two regions may share a common evolutionary origin, and that RA may have evolved as a specialized expansion of AId (*Feenders et al., 2008*) that sends heavily myelinated fibers to brainstem vocal-motor neurons for fast-spike signaling (*Figures 1C and 2B*; *Alcami and El Hady, 2019*).

In conclusion, RAPNs in zebra finches exhibit many fundamental molecular and functional similarities to primate Betz cells, that may be involved in fine digit movements (*Lemon and Kraskov, 2019*; *Tomasevic et al., 2022*). In combination with their well-defined role in controling singing behaviors, this study identifies RAPNs as a novel and more accessible model for studying the properties of Betz-like pyramidal neurons that offer the temporal precision required for complex learned motor behaviors.

## Methods

### Animal subjects

All of the work described in this study was approved by OHSU's Institutional Animal Care and Use Committee (Protocol #: IP0000146) and is in accordance with NIH guidelines. Zebra finches (*Taeniopygia guttata*) were obtained from our own breeding colony. All birds used were male and >120 days post hatch. Birds were sacrificed by decapitation and their brains removed. For electrophysiology experiments brains were bisected along the midline, immersed in ice-cold cutting solution, and processed as described below. For in situ hybridization experiments brains were cut anterior to the tectum and placed in a plastic mold, covered with ice-cold Tissue-Tek OCT (Sakura-Finetek; Torrance, CA, USA), and frozen in a dry ice/isopropanol slurry and processed as described below.

### In situ hybridization

To compare mRNA expression levels for *KCNC1, KCNC2, KCNC3, KCNC4, KCNMA1, KCNQ2, KCNQ3, KCNA1, KCNA2, and KCNA6* across RA and AId, brains sections (thickness = 10 μm) were cut coronally on a cryostat and mounted onto glass microscope slides (Superfrost Plus; Fisher Scientific, Hampton, NH, USA), briefly fixed, and stored at –80°C. For each brain, every 10th slide was fixed and stained for Nissl using an established cresyl violet protocol. Slides were examined under a bright-field microscope to identify sections containing the core region of RA and AId as previously defined (*Nevue et al., 2020*). In situ hybridization was conducted using an established protocol (*Carleton et al., 2014*). Briefly, slides were hybridized under pre-optimized conditions with DIG-labeled ribo-probes synthesized from BSSHII-digested cDNA clones obtained from the ESTIMA: songbird clone

collection (*Replogle et al., 2008*). Specific clones corresponded to GenBank IDs CK302978 (*KCNC1*; Kv3.1), DV951094 (*KCNC2*; Kv3.2), DV953393 (*KCNC3*; Kv3.3), CK308792 (*KCNC4*; Kv3.4), DV954467 (*KCNMA1*; BK), FE737967 (*KCNA1*; Kv1.1), FE720882 (*KCNA2*; Kv1.2), FE733881 (*KCNA6*; Kv1.6), DV954380 (*KCNQ2*; Kv7.2), and CK316820 (*KCNQ3*, Kv7.2). After overnight hybridization, slides were washed, blocked, incubated with alkaline phosphatase conjugated anti-DIG antibody (1:600; Roche, Basal, Switzerland) and developed overnight in BCIP/NBT chromogen (Perkin Elmer; Waltham, MA, USA). Slides were coverslipped with VectaMount (Vector, Newark, CA, USA) permanent mounting medium, and then digitally photographed at 10× under bright-field illumination with a Lumina HR camera mounted on a Nikon E600 microscope using standardized filter and camera settings. Images were stored as TIFF files and analyzed further using the FIJI distribution of ImageJ (*Schindelin et al., 2012*). We note that high-resolution parasagittal images depicting expression of *KCNC1*, *KCNC2*, *KCNA1*, *KCNA6*, *KCNQ2,* and *KCNMA1* in RA of adult male zebra finches are available on the Zebra Finch Expression Brain Expression Atlas (ZEBrA; https://www.zebrafinchatlas.org). All probes were evaluated for specificity by examining their alignment to the zebra finch genome using BLAST (as detailed previously, *Lovell et al., 2020*); all probes used were verified to align specifically to the target locus, with no significant secondary alignments. Also, importantly, our in situs are run under optimized conditions that result in no detectable signal upon omission of probe or the anti-DIG antibody.

For each gene, we quantified both expression levels based on labeling intensity (i.e. average pixel intensity) and the number of cells expressing mRNA per unit area. We measured the average pixel intensity (scale: 0–256) in a 200×200 $\mu m^2$ window placed over each target area in the images of hybridized sections. To normalize signal from background we subtracted an average background level measured over an adjacent control area in the intermediate arcopallium that was deemed to have no mRNA expression. The expression ratio was calculated as $RA_{OD}/AId_{OD}$ where values greater than 1 are more highly expressed in RA and values less than 1 are more highly expressed in AId. We also quantified the number of labeled cells in each arcopallial region by first establishing a threshold of expression 2.5× above the background level. Standard binary filters were applied and the FIJI 'Analyze Particles' algorithm was used to count the number of labeled cells per 200 $\mu m^2$.

## Slice preparation for electrophysiology experiments

Frontal (180 $\mu m$ for current clamp and 150 $\mu m$ for voltage clamp) slices were cut on a vibratome slicer (VT1000, Leica) in an ice-cold cutting solution containing (in mM): 119 NaCl, 2.5 KCl, 8 $MgSO_4$, 16.2 $NaHCO_3$, 10 HEPES, 1 $NaH_2PO_4$, 0.5 $CaCl_2$, 11 D-glucose, 35 sucrose pH 7.3–7.4 when bubbled with carbogen (95% $O_2$, 5% $CO_2$; osmolarity ~330–340 mOsm). Slices were then transferred to an incubation chamber containing artificial cerebral spinal fluid (aCSF) with (in mM): 119 NaCl, 2.5 KCl, 1.3 $MgSO_4$, 26.2 $NaHCO_3$, 1 $NaH_2PO_4$, 1.5 $CaCl_2$, 11 D-glucose, 35 sucrose pH 7.3–7.4 when bubbled with carbogen (95% $O_2$, 5% $CO_2$; osmolarity ~330–340 mOsm) for 10 min at 37°C, followed by a room temperature incubation for ~30 min prior to start of electrophysiology experiments.

## Patch-clamp electrophysiology

RA and AId could be readily visualized in via infra-red differential interference contrast microscopy (IR-DIC) (*Figure 2B*). Whole-cell patch-clamp recordings were performed at room temperature (~24°C) unless otherwise indicated. For experiments performed at 40°C, the bath solution was warmed using an in-line heater (Warner Instruments, Hamden, CT, USA). The temperature for these experiments varied up to ±2°C.

Slices were perfused with carbogen-bubbled aCSF (1–2 ml/min) and neurons were visualized with an IR-DIC microscope (Zeiss Examiner.A1) under a 40× water immersion lens coupled to a CCD camera (Q-Click; Q-imaging, Surrey, BC, Canada). Whole-cell voltage- and current-clamp recordings were made using a HEKA EPC-10/2 amplifier controlled by Patchmaster software (HEKA, Ludwigshafen/Rhein, Germany). Data were acquired at 100 kHz and low-pass filtered at 2.9 kHz. Patch pipettes were pulled from standard borosilicate capillary glass (WPI, Sarasota, FL, USA) with a P97 puller (Sutter Instruments, Novato, CA, USA). All recording pipettes had a 3.0–6.0 M$\Omega$ open-tip resistance in the bath solution. Electrophysiology data were analyzed off-line using custom written routines in IGOR Pro (WaveMetrics, Lake Oswego, OR, USA).

For current-clamp recordings, intracellular solutions contained (in mM): 142.5 K-gluconate, 21.9 KCl, 5.5 $Na_2$-phosphocreatine, 10.9 HEPES, 5.5 EGTA, 4.2 Mg-ATP, and 0.545 GTP, pH adjusted to 7.3

with KOH, ~330–340 mOsm. Synaptic currents were blocked by bath applying Picrotoxin (100 µM), DL-APV (100 µM), and CNQX (10 µM) (Tocris Bioscience) for ~3 min prior to all recordings. To initiate current-clamp recordings, we first established a giga-ohm seal in the voltage-clamp configuration, set the pipette capacitance compensation (C-fast), and then set the voltage command to –70 mV. We then applied negative pressure to break into the cell. Once stable, we switched to the current-clamp configuration. Experiments in current clamp were carried out within a 15 min period. AP half-width was defined as the width of the AP half-way between threshold (when the rate of depolarization reaches 10 V/s) and the AP peak. The maximum depolarization and repolarization rates were obtained from phase plane plots generated from averaged spontaneous APs. We noted that the resting membrane potential tended to hyperpolarize to the same degree (~10 mV) in both RA and AId after positive current injections during these current-clamp recordings (*Alexander et al., 2019*). Recordings in which the resting membrane potential deviated by >10 mV were discarded. We note that recordings were not corrected for a calculated liquid junction potential of +9 mV.

Estimated current-clamp measurements of membrane capacitance ($C_m$) was obtained by dividing the time constant ($\tau_m$; fit with a single or double exponential, where a weighted average was derived using the relative amplitudes, at the onset of a negative current injection) with the input resistance ($R_{in}$; calculated slope of V-I plot):

$$\tau_m = R_{in} \cdot C_m$$

For RAPNs at room temperature the $\tau_m$ was well fit by a single exponential in 62.5% of neurons and was fit better by a double exponential in 37.5% of the neurons with a slow component contributing 43.6% of the amplitude. At high temperature the $\tau_m$ was well fit by a single exponential in 63.6% of neurons and was fit better by a double exponential in 36.4% of the neurons with a slow component contributing 50.2% of the amplitude. Our $\tau_m$, $R_{in}$, and $C_m$ values agree well with previous estimates from room temperature recordings (*Zemel et al., 2021*). For AId neurons at room temperature the $\tau_m$ was well fit by a single exponential in 50% of neurons and by a double exponential fit in 50% with a slow component contributing 73.8% of the amplitude. At high temperature the $\tau_m$ was well fit by a single exponential in 41.7% of neurons and was fit better by a double exponential in 58.3% of the neurons with a slow component contributing 47.8% of the amplitude. The values of $\tau_m$ and $R_{in}$ were obtained from initial onset to peak voltage change since steady-state values at the end of the current injection activate $I_h$ currents, especially at higher temperatures (*Zemel et al., 2021*). For voltage-clamp recordings, we attempted to limit the voltage and space clamp error by (1) cutting thinner slices (~150 µm) to eliminate more processes, (2) decreased the intracellular K⁺ concentration to decrease the driving force, and (3) compensated the series resistance electronically to 1 MΩ. The intracellular solutions contained the following (in mM): 75 K-gluconate, 5.5 Na₂-phosphocreatine, 10.9 HEPES, 5.5 EGTA, 4.2 Mg-ATP, and 0.545 GTP, pH adjusted to 7.3 with KOH, adjusted to ~330–340 mOsm with sucrose. $R_s$ was compensated to 1 MΩ, the uncompensated $R_s$ = 7.7 ± 0.4 MΩ (mean ±SE; N=12 RAPNs and AId neurons). We ensured exclusion of interneurons by briefly observing the AP waveform in current clamp prior to switching to voltage clamp (*Figure 2—figure supplement 1*; *Zemel et al., 2021*). In order to isolate K⁺ currents, slices were exposed to bath applied CdCl₂ (100 µM), TTX (1 µM), Picrotoxin (100 µM), CNQX (10 µM), and APV (100 µM) for ~5 min prior to running voltage-clamp protocols. After protocols were applied, TEA (500 µM) was bath applied and the same voltage-clamp protocols were repeated after K⁺ currents were eliminated. K⁺ currents were isolated by subtracting the TEA-insensitive current traces from the initial traces. Capacitive currents generated during voltage-clamp recordings were eliminated by P/4 subtraction. Recordings were not corrected for a measured liquid junction potential of +12 mV.

Within RA, inhibitory GABAergic interneurons are sparse, and are easily distinguished from RAPNs based on differences in their firing properties (*Miller et al., 2017*; *Spiro et al., 1999*; *Zemel et al., 2021*). In situ hybridization for GAD1 and -2 indicates low proportions of GABAergic cells in AId as well (*Lovell et al., 2020*; *Pinaud and Mello, 2007*). Accordingly, in our AId recordings we very infrequently encountered neurons with properties resembling interneurons (*Figure 2—figure supplement 1*; *Garst-Orozco et al., 2014*; *Kittelberger and Mooney, 1999*; *Liao et al., 2011*; *Miller et al., 2017*; *Spiro et al., 1999*). Because we were primarily interested in recording from excitatory neurons, and those represented the vast majority of recorded cells based on electrophysiological criteria, we excluded putative GABAergic interneurons in AId from further analysis.

## Morphology

For these experiments Biocytin (4 mg/ml; Sigma, St. Louis, MO, USA) and Alexa Fluor 488 (Life Technologies, Carlsbad, CA, USA) were included in the intracellular solution used in current-clamp experiments. Upon entering the whole-cell current-clamp configuration, the current was set to 0 pA and the cell was held in for 20–30 min at room temperature to allow for complete filling. The electrode was then slowly removed at a diagonal angle as the fluorescence was monitored to determine when the electrode detached. Upon complete separation of the electrode from the cell, the slice was placed in 4% paraformaldehyde overnight. The slice was then washed in PBS with 0.25% Triton X-100 (3× for 10 min), blocked in a PBS solution containing 1% skim milk (1 hr) and then stained in a PBS solution containing 1% skim milk and avidin conjugated to Alexa Fluor 594 (1:200, Life Technologies, Carlsbad, CA, USA) for 2 hr. The slice was then washed in PBS with 0.25% Triton X-100 (3× for 10 min) before a final wash in PBS followed by mounting on a cover slip. Images were captured with a Zeiss LSM 980 Airyscan 2 confocal microscope.

Maximum projections of the images were made and analyzed in FIJI-ImageJ (*Schindelin et al., 2012*) to calculate the 2D soma area, dendritic complexity (*Sholl, 1956a*), and estimate spine density (as measured from averaged 30 μm stretches from multiple tertiary branches in each cell) shown in *Figure 1*. In a subset of these neurons (three RAPNs and two AId neurons) we used the recently developed, open-source software, ShuTu (*Jin et al., 2019*), to reconstruct the morphologies in 3D (see examples in *Figure 1—figure supplement 1A–B*). 3D renderings were expanded twofold in the Z-direction to allow for better visualization of cellular processes. Despite the planar appearance of reconstructed neurons, previous imaging work in RAPNs and hippocampal neurons suggests these dendrites tile equally across planes (*Spiro et al., 1999*) and have been disproportionately compressed in the Z-axis as a result of tissue processing (*Pyapali et al., 1998*). We used ShuTu's automatic reconstruction script to trace high-contrast neurites filled with biocytin and to estimate their diameters. For low-contrast, broken or occluded processes, we manually corrected the reconstruction in ShuTu's GUI. The reconstructions were stored in SWC format. We used ShuTu's GUI to manually trace all visible dendritic spines. We annotated spines with thin necks and mushroom-like shapes using single SWC points projected away from the dendrite and following the spine's neck direction. For spines with curved necks, irregular shapes or filopodia-like spines, we used multiple SWC points tracing their entire extent. We estimated spine areas in two ways. We treated spines annotated by a single SWC point as spheres attached by cylindrical necks with an average radius of 0.5 pixels (0.066 μm). For spines with multiple SWC points, we used the trapezoidal cylinders defined by every pair of consecutive SWC point. We also added a dome cap to the last SWC point. To obtain a continuous profile of spine density along each dendritic branch (*Figure 1—figure supplement 1E*), we used a rolling window of width 20 SWC points (window length fluctuated between 5 μm and 39 μm, with an average of 17 μm for all cells) and of step size 1 SWC point. From the continuous spine density profile, we computed the average spine density of each dendritic segment (*Figure 1—figure supplement 1C*). We excluded from the density estimation of *Figure 1—figure supplement 1C* segments branching off the soma with fewer than 10 spines and any segment shorter than 20 μm. We estimated the total dendritic area and length of each cell (*Figure 1*, *Supplementary file 1*) from the reconstructions by constructing trapezoidal cylinders for each pair of consecutive SWC point. Axonal area and length were estimated in the same way, although we were only able to connect a small fraction of axon cable to the soma for three neurons. For two neurons we were not able to trace any portion of the axon (*Figure 1*, *Supplementary file 1*).

We estimated the surface area of the soma of each neuron by triangulating its 3D structure. We first used ShuTu's GUI (*Jin et al., 2019*) to manually trace binary masks of the somas in all slices. Next, we traced the contours of the masks with edge detecting filters. We then took a fixed fraction of 20 equally spaced points in each contour in order to generate polygonal contours. All polygonal contours in the end had the same number of edges. Next, we applied a three-point moving average filter along each sequence of vertex in the Z-direction to smooth out rough edges. We then calculated the areas of the triangles formed by the edges and vertices of each adjacent pair of polygonal contours. We added to this number the surface areas of the first and last non-zero binary masks to represent the caps. We also corrected for the areas of the surface patches where the dendrites attach to the soma. Since these surface patches are small, we approximated them by the cross-sectional area of the first dendrite SWC point connected to the soma.

We estimated the volumes of the dendrites using 3D binary masks which we constructed from the SWC structure. First, we re-scaled the SWC along the Z-axis to match the scale in the XY plane. Next, we interpolated the space between each connected pair of SWC points by trapezoidal cylinders with radii equal to the SWC points' radii. The binary mask assigns ones to voxels intersecting either an SWC point or a trapezoidal cylinder and zeros otherwise. The volume of the dendrite is the fraction of active voxels in the mask multiplied by the volume of the voxel.

We estimated the volumes of the somas using an adaptation of our method for the surface areas. Starting from the polygonal contours representing the Z-stack, we used the polygons' vertices and baricenters to split the soma's volume into a collection of tetrahedrons. The vertices of an edge in a slice, the vertices of the closest edge in the next slice, and the baricenters of the two contours define four tetrahedrons for which the volume can be easily calculated. The total volume of the soma is the sum of the volumes of the individual tetrahedrons.

We approximated the volumes of the spines directly from the SWC structure. Since the spines are small, we estimated their volumes to be approximately the sum of the volumes of the individual SWC points.

## Comparative genomics of the *KCNC*/Kv3 (Shaw-related) potassium channel genes

To identify the full set of genes comprising this gene family in zebra finches, we first retrieved all genes annotated as *KCNCs* (voltage-gated channel subfamily C members) from the latest RefSeq database for zebra finch (Annotation Release 106; GCF_003957565.2 assembly). We next retrieved the similarly annotated genes in other selected songbird and non-songbird avian species, observing the immediate synteny and correct cross-species BLAST alignments. To verify the correct orthology of avian genes to corresponding members of this gene family in mammals, we conducted cross-species BLAST searches, noting the top-scoring reciprocal cross-species alignments as well as conserved synteny as orthology criteria. Cases where the avian gene prediction was located on a small sized scaffold with no synteny information were not included in our analysis, as rigorous confirmation of orthology was not possible. For the *KCNC1L* analysis (*Figure 7—figure supplement 1B*), we only examined songbird species with no sequence gaps in the genomic region of interest. To build cladistic trees for evolutionary inferences, we also included (as outgroups to birds and mammals) representative extant organisms from selected branches of major vertebrate groups, as appropriate, including non-avian sauropsids (crocodiles, turtles, lizards), amphibians, and bony fishes where the genes of interest have been identified, and gene orthology to species with more complete and annotated genomes (e.g. human, mouse) can be clearly established.

## Pharmacological compounds

DL-APV, Picrotoxin, CNQX, XE991, and iberiotoxin were purchased from Tocris Biosciences (Bristol, UK). TEA and 4-AP were purchased from Sigma-Aldrich (St. Louis, MO, USA). α-DTX was purchased from Alomone labs (Jerusalem, Israel). AUT5 was provided as a gift from Autifony Therapeutics (Stevenage, UK). For all experiments synaptic blockers were applied for approximately 2 min prior to baseline recordings. Kv antagonists and AUT5 were applied for approximately 3 min prior to subsequent recordings. Recordings during washout experiments were performed in 3 min intervals.

## Statistical data analysis and curve fitting

Data were analyzed off-line using IgorPro software (WaveMetrics). Statistical analyses were performed using Prism 4.0 (GraphPad). Specific statistical tests and outcomes for each analysis performed are indicated in the respective Figure Legends and Tables. Means and SE are reported, unless otherwise noted. Electrophysiology and morphology data is described as technical replicates for individual cells while molecular data is described as biological replicates for individual birds.

## Acknowledgements

We thank Autifony Therapeutics and Dr. Martin Gunthorpe for providing the AUT5 compound for this study, Drs. Manuel Covarrubias and Qiansheng Liang for thoughtful comments on experimental design using AUT5, Dr. Pepe Alcami for discussions on RA projection neurons and Dr. Cesar Ceballos

for his technical recommendations regarding biocytin cell fills. Grants from NIH and NSF to CVM and HvG.

## Additional information

### Competing interests

Henrique von Gersdorff: Reviewing editor, eLife. The other authors declare that no competing interests exist.

### Funding

| Funder | Grant reference number | Author |
| --- | --- | --- |
| National Science Foundation | NSF1456302 | Claudio V Mello |
| National Science Foundation | NSF1645199 | Claudio V Mello |
| National Institutes of Health | GM120464 | Claudio V Mello |
| National Institutes of Health | DC004274 | Henrique von Gersdorff |
| National Institutes of Health | DC012938 | Henrique von Gersdorff |
| National Institutes of Health | AG055378 | Benjamin M Zemel |
| National Science Foundation | NSF2154646 | Claudio V Mello |
| National Science Foundation | NSF2154646 | Henrique von Gersdorff |

The funders had no role in study design, data collection and interpretation, or the decision to submit the work for publication.

### Author contributions

Benjamin M Zemel, Conceptualization, Data curation, Formal analysis, Funding acquisition, Investigation, Methodology, Resources, Software, Validation, Visualization, Writing – original draft, Writing – review and editing, designed electrophysiology and morphology experiments; Alexander A Nevue, Conceptualization, Formal analysis, Investigation, Methodology, Resources, Visualization, Writing – review and editing, designed molecular biology experiments and genomic analysis; Leonardo ES Tavares, Conceptualization, Software, Formal analysis, Investigation, Visualization, Methodology, Writing – review and editing, designed and carried out morphometric analyses using the ShuTu software package; Andre Dagostin, Investigation, Visualization, Methodology, acquired electrophysiology data and performed biocytin fills; Peter V Lovell, Conceptualization, Funding acquisition, integrated molecular and electrophysiological data; Dezhe Z Jin, Conceptualization, Resources, Software, Formal analysis, Supervision, Investigation, Visualization, Methodology, Writing – review and editing, designed and managed morphometric analyses using the ShuTu software package; Claudio V Mello, Conceptualization, Resources, Formal analysis, Supervision, Funding acquisition, Investigation, Visualization, Methodology, Project administration, Writing – review and editing, designed molecular biology experiments and genomic analysis; Henrique von Gersdorff, Conceptualization, Resources, Formal analysis, Supervision, Funding acquisition, Visualization, Methodology, Project administration, Writing – review and editing, designed electrophysiology and morphology experiments

### Author ORCIDs

Benjamin M Zemel ⓘ http://orcid.org/0000-0002-6701-0647
Leonardo ES Tavares ⓘ http://orcid.org/0000-0002-8642-5186
Claudio V Mello ⓘ http://orcid.org/0000-0002-9826-8421

Henrique von Gersdorff ![ORCID] http://orcid.org/0000-0002-4404-3307

### Ethics

This study was performed in strict accordance with the recommendations in the Guide for the Care and Use of Laboratory Animals of the National Institutes of Health. All of the animals were handled according to approved institutional animal care and use committee (IACUC) protocols of the OHSU (IACUC # IP0000146).

### Decision letter and Author response

Decision letter https://doi.org/10.7554/eLife.81992.sa1
Author response https://doi.org/10.7554/eLife.81992.sa2

---

## Additional files

### Supplementary files

• Supplementary file 1. Raw data from all figure panels and tables. Geometrical properties and spine densities measured from the reconstructions of the neuronal morphologies. Values of the axon area include only the visible portions connected to the soma. Axons excluded from volume measurements.

• MDAR checklist

• Source data 1. Raw data from all figure panels and tables.

### Data availability

A source Excel data file is included with the manuscript and a raw data file has been uploaded to the DRYAD data base (DOI https://doi.org/10.5061/dryad.1zcrjdfvs).

The following dataset was generated:

| Author(s) | Year | Dataset title | Dataset URL | Database and Identifier |
|---|---|---|---|---|
| Zemel BM, Nevue AA, Tavares LES, Dagostin A, Lovell PV, Jin DZ, Mello CV, von Gersdorff H | 2023 | Data from Motor Cortex Analogue Neurons in Songbirds Utilize Kv3 Channels to Generate Ultranarrow Spikes | https://doi.org/10.5061/dryad.1zcrjdfvs | Dryad Digital Repository, 10.5061/dryad.1zcrjdfvs |

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
