## [Editor Report]

This paper carries significant novelty, as it describes a mechanism for fast information processing in the innovative zebra finch model and provides strong experimental support for the fundamental cellular properties underlying such high-frequency signaling.

---

## [Decision Letter]

**Decision letter after peer review:**

Thank you for submitting your article "Cortical Betz cells analogue in songbirds utilizes Kv3.1 to generate ultranarrow spikes" for consideration by *eLife*. Your article has been reviewed by 3 peer reviewers, one of whom is a member of our Board of Reviewing Editors, and the evaluation has been overseen by John Huguenard as the Senior Editor. The following individual involved in the review of your submission has agreed to reveal their identity: Ian D Forsythe (Reviewer #1).

Essential revisions:

1) The correspondence between RA cells and cortical Betz cells is a stretch.

This seems more of a discussion point than an essential part of the title, abstract, or introduction.

2) The evidence of a specific role for Kv3.1 is not conclusive and relevant claims should be moderated.

3) A valuable pharmacological approach is used. Evidence of stability of responses and reversibility of drug effects should be provided for each of the major findings.

*Reviewer #1 (Recommendations for the authors):*

This is an extensive piece of work and it contains some good quality data, but the great breadth of the article means that many elements of the results are too superficial and/or lack sufficient detail to deal with certain aspects and complexities of Kv3 physiology. There are also some problems in the interpretation of the voltage-clamp data which together weaken the overall findings.

1). Page 1. The first paragraph of the introduction over-emphasises a link to mammalian Betz cells; this would seem better justified as a discussion point. Perhaps you feel the need to justify the avian system to mammalian-biased readers, but the lack of transgenic zebrafinch models undermines the strength of your arguments for doing this work in the bird.

2) Page 3. Morphology. While the detail and work conducted in contrasting the morphology of RA and AId neurons are appreciated, your cell numbers are low (only 2 AId neurons) and this reduces the confidence in the observed differences. Also, many of the parameters are presented as % differences – this would be better presented as different absolute values (these are often only shown on the figure graph and are difficult to estimate). It is not clear how the observed differences relate to the Kv3 question, other than in establishing some convergence with mouse cortical neurons. This could be a distinct study in itself but is not a good fit for this manuscript.

3) Page 4 ln 179. Electrophysiology. The data is usefully presented in absolute terms in Table 1, but the differences should be described in the same terms, not as % (as in ln192) nor as qualitative comments (as in lns 198 – 203).

4) Page 5, ln 208-217. This is a methodological aside and is not relevant except to demonstrate the care that the authors have taken to exclude inhibitory neurons from their data set.

5) Page 5, ln 230-254. The data is consistent with a larger effect of TEA and 4-AP on RA repolarization. Again, absolute measures of duration rather than % change should be quoted and compared for the data in figure 4. This will be important if anyone were to model these conductances and neuronal properties in the future. APs in the Ald neurons are already longer, so a similar effect of TEA or 4AP might be expected to be proportionally smaller (if presented as %).

Figure 4 clearly shows that TEA and 4-AP have direct effects on the AP of RA cells, but less so on Aid cells. But you can only infer that this is mediated by an action on Kv3 channels. The data is very difficult to interpret as all the numerical graphs are presented in fold change in AP halfwidth. How long were the drugs applied, and was recovery data collected to demonstrate washout? It is a pedantic point, but have you checked that the pharmacological evidence cited (from mammals) applies to birds? Since you show no effect in Supl Figure 4, a key issue would be: does α-DTx or iberiotoxin block Kv1 and BK channels (respectively) in the bird; have you positive controls or can you provide citations of such (or at least checked for precise homology at the respective toxin binding sites)?

6) The effect of TEA and 4AP differs in figure 5 (first AP) from figure 4 (AP train). The effect on the half-width of the 1st AP seems smaller than on subsequent APs in the train – particularly noticeable in 5A and 5B. What is the explanation for this (see point 8 below)?

7) The AP duration is also changing during the trains in Supl Figure 5. There are a few minor inconsistencies here. For example, XE991 has a significant effect in the lowest current injections in S5c. The control peak firing rates (spikes/sec) in S5E are lower (~70 vs ~100) than in S5A and S5C.

8) Page 6 and figure 6. Voltage-clamp data is well presented and clearly shows larger outward currents in the RA neurons, with little or no 'low-voltage-activated' (steps to -30 mV; potential DTx-sensitive current) in either neuron type. The 'high-voltage activated' currents are very slow activating… And so would be hardly activated during a single fast AP lasting less than a ms….. This undermines your arguments that this is Kv3 on two counts: it is too slow to activate (taking over 100ms) and only differs by around 25% between RA and AId neurons. The key current for the AP repolarization, especially for the first AP in a train, is most likely to be the fast inactivating current. From the evidence in figure 6, it is not clear that this is a classic A-current (Kv4), because no estimation of steady-state inactivation has been conducted.

Although this undermines your current hypothesis that this is a Kv3.1 current, the possibility that the Kv3 current is fast inactivating (i.e. a Kv3.3 or Kv3.4-like current) is very interesting. In Figure 6G, you appear to show that both the inactivating and slow-activating outward currents are TEA-sensitive. Because the activation of the slow outward current overlaps with the decay of the fast transient outward current, any estimation of one will be contaminated by the other. Ideally, you could steady-state inactivate the transient current so you can measure the slow-activating current. This would then allow you to estimate the 'contamination' of one current by the other.

9) P7 ln 335. Another paper has described kcnc3/Kv3.3 in a seabird (De Paoli-Iseppi et al. PLOS One 12: e0189181, 2017).

10) Figure 7 and Page 7/8. The heading on ln 319 states, that Kv3.1 is the TEA-sensitive ion channel subunit… But this seems a rather weak point and could be disputed, as you have measured mRNA, not protein (and the interpretation of the current is ambiguous). The in situ hybridisation for each of the Kv3 genes is interesting, but the quantification is insufficient. The lack of absolute values (or at least relative to some 'housekeeping' gene) for the mRNA expression is a serious weakness. The ratios expressed here between two nuclei in Figure 7 hide/obscure important information and have contributed to the misinterpretation of the physiology. For example, one could argue on the basis of Fig7A that all Kv3 genes are present and that kcnc3 is most striking in that it highlights the two nuclei of interest, is present in both and you have the positive control of Purkinje neurons. Fig7B and 7C are best omitted… The expression ratio of RA to Ald has little or no relevance to a measurable parameter in your study. Perhaps you could argue some relevance if you had done a more quantitative measure and/or western blotting, but this has not been done. Your own evidence clearly indicates that multiple Kv3 subunits are present (Figure 7A), but you have no measure of the efficacy with which the mRNA is translated into protein (it is also hard to know what constitutes a 'background' value for any of the mRNA probes).

11. AUT5. This is not a well-designed experiment, since you apply a dose that is half-maximal (1uM when the EC50 is 1.3uM: ln 391), so you would be unlikely to observe more than half of the potential effect. For most readers, this comes across as a rather minor point and I think they will wonder why they need to know this information, certainly it is a weak figure with which to finish. Perhaps if AUT5 was shown to change the finch song or other behaviour in the opposite way to blocking the channel it would be interesting. AUT5 effects have also been noted previously in other publications:

2015. Biophysical characterization of KV3.1 potassium channel activating compounds. Taskin B, von Schoubye NL, Sheykhzade M, Bastlund JF, Grunnet M, Jespersen T.

Eur J Pharmacol. 2758:164-70.

2017. Kv3.1/Kv3.2 channel positive modulators enable faster activating kinetics and increase firing frequency in fast-spiking GABAergic interneurons. Boddum K, Hougaard C, Xiao-Ying Lin J, von Schoubye NL, Jensen HS, Grunnet M, Jespersen T. Neuropharmacology, 118:102-112.

*Reviewer #3 (Recommendations for the authors):*

This paper was a pleasure to read and goes through a logical sequence of experiments to pinpoint the role of Kv3(.1?) in the fast-spiking of RAPN neurons. There are large replicate numbers for almost all findings, and admirably, there is confirmation of the main result at physiological temperature. The phase plots in Figure 4 make clear that 4AP and TEA, at the concentrations used, have little effect on Ald neurons, but strong effects on RAPNs in both current clamp and voltage clamp. The methods developed to improve voltage clamp compliance are to be commended.

1) The paper has an unclear focus. In some ways, it focuses on the properties of RAPN neurons and how they might generate fast APs. Yet in other contexts, the paper seems to center on the description of a previously uncharacterized cell type, the neurons of Ald.

2) The pharmacological analysis of repetitive spike firing is lacking. For example, the traces in Figure 5B and 5D, and 8B and 8D, DO in fact show effects of 4AP/TEA/AUT5 on Ald neurons. This raised two issues. A) Repeated measures, such as a paired t-test I (before and after the drug, on a per cell basis) are arguably more appropriate than group averages at each current injection strength. B) Some effort should be dedicated to reversibility. While it can be technically challenging to hold a cell for the total time required to completely wash out the drug, at least a few examples should be provided to confirm that "drug" effects are not due to time-dependent changes in cell physiology.

3) For fast repolarization, it would likely be better to focus mainly on the early current, highlight with the red bars in figure 6, rather than on the current persisting until 200 ms.

4) There is a bit of circular logic regarding the description of the effects of AUT5 on spike depolarization (Figure 8A). If RA affects the maximum depolarization, then this could secondarily increase the recruitment of K channels and in turn produce a faster rate of repolarization.

5) While voltage clamp compliance is likely much improved by the approaches taken, it will still not be perfect, and there will remain issues of non-isopotentiality. This should be discussed.

6) The correspondence to Betz cells is oversold and should be removed from the title and abstract. There are many differences, as the authors note. A) The RA cells are not necessarily larger than other motor output neurons as with Betz cells, B) They have presumably developed along a unique evolutionary path, C) Other central features of Betz cells (conduction velocities, myelination, axon fiber diameter) are not reported.

---

## [Author Response]

Essential revisions:1) The correspondence between RA cells and cortical Betz cells is a stretch.This seems more of a discussion point than an essential part of the title, abstract, or introduction.

We agree with this request, as we may have initially overemphasized the analogy between RA projection neurons and Betz cells. We have now (1) removed the comparison with Betz cells from the Title, (2) tempered our language in the Abstract, where we now clarify that RA projection neurons and Betz cells share some physiological and molecular properties, and (3) removed the opening paragraph in the Introduction that discusses mammalian Betz cells, in favor of the second paragraph introducing the zebra finch model to study aspects of fine, fast motor control. As also suggested by the reviewer, we expand the comparison between RA projection neurons and Betz cells in the Discussion, noting that we also disclose several differences between these cell types.

2) The evidence of a specific role for Kv3.1 is not conclusive and relevant claims should be moderated.

We agree that we cannot conclude from our data that Kv3.1 is the sole voltage-gated K^+^ channel responsible for AP repolarization in RA projection neurons. Our title has changed now to mention “Kv3 subunits”. However, we have found marked contrasts in excitable properties (e.g. spike half-width, maximum repolarization rate) between upper motor neurons in RA and AId, and thus identified physiological specializations characteristic of just RA projection neurons (RAPNs). The molecular data also give clear indication that Kv3.1 is differentially expressed between RAPNs and AId neurons, in contrast to the other TEA-sensitive Kv3 family members, which are not differentially expressed between RA and AId. Coupled with the data from the Kv3.1/Kv3.2 specific agonist, AUT5, we argue that Kv3.1 is the most likely subunit to explain the difference in spike properties between these two regions. Importantly, our paper shows the presence of Kv3.3 in RA and AId and also possible functional evidence of Kv3.4, since we show that a component of the K currents inactivates very quickly (see Figure 6A and 6G).

To address this important issue, we have changed the wording throughout the manuscript to moderate our claims about a unique role of Kv3.1 in RA. This includes changes in the Title (line 1), Abstract (line 31-35), Results (line 395-399, 409-410, 446), Figure Legends (line 1592, 1624) and Discussion (line 546, 551-553). We have also added a new paragraph at the beginning of Results (lines 105-116) to clarify the rationale for our regional comparative approach.

3) A valuable pharmacological approach is used. Evidence of stability of responses and reversibility of drug effects should be provided for each of the major findings.

We show evidence of stability and reversibility of drug effects in an added new figure (Figure 5—figure supplement 1), and describe the washout of TEA, 4-AP and AUT5 in a related passage of Results (lines 262-271, 442-444), and provide further details in Methods (lines 894-897). We note that the opposing effects of AUT5 versus 4-AP and TEA, as well as the stability of recordings during administration of iberiotoxin already provide an indication of stable recordings that can be differentially affected depending on the types of compounds applied to our slices, but we agree that the additional requested data help to solidify the study.

Reviewer #1 (Recommendations for the authors):This is an extensive piece of work and it contains some good quality data, but the great breadth of the article means that many elements of the results are too superficial and/or lack sufficient detail to deal with certain aspects and complexities of Kv3 physiology. There are also some problems in the interpretation of the voltage-clamp data which together weaken the overall findings.1) Page 1. The first paragraph of the introduction over-emphasises a link to mammalian Betz cells; this would seem better justified as a discussion point. Perhaps you feel the need to justify the avian system to mammalian-biased readers, but the lack of transgenic zebrafinch models undermines the strength of your arguments for doing this work in the bird.

We thank the reviewer for the detailed and helpful comments. We agree with the assessment that we overemphasized the similarities to mammalian Betz cells, and have now eliminated the opening paragraph of the Introduction, while reserving that topic for the Discussion. The Introduction now starts with the paragraph where we introduce zebra finches as a useful model for studying regulation of intrinsic neuronal excitability in the context of refined, fast motor control. In that regard, we note that more traditional genetic models like mice lack the equivalent of a cortical vocal nucleus like RA and its specialized projection neurons, as well as the vocal learning behavior this nucleus subserves. Thus, in spite of the current limitations in applying transgenic methods to these birds, their use provides a unique opportunity to study features of fine motor control systems that are uniquely shared with primates and not apparent in rodents.

2) Page 3. Morphology. While the detail and work conducted in contrasting the morphology of RA and AId neurons are appreciated, your cell numbers are low (only 2 AId neurons) and this reduces the confidence in the observed differences. Also, many of the parameters are presented as % differences – this would be better presented as different absolute values (these are often only shown on the figure graph and are difficult to estimate). It is not clear how the observed differences relate to the Kv3 question, other than in establishing some convergence with mouse cortical neurons. This could be a distinct study in itself but is not a good fit for this manuscript.

The reviewer brings up relevant points about the cell morphology analysis. While we agree that the cell morphology analyses we presented do not directly relate to the Kv3-related experiments, ours is a first characterization of neuronal morphology in AId, and how it compares to RAPN morphology. Importantly, while the N of the elaborate ShuTu analysis was small, it was meant to complement the findings from our more extensive FIJI analysis (N=6 per cell-type). The ShuTu analysis confirmed, using a different approach, the findings of cell type differences in spine density and dendrite area. To fully address the reviewer’s request, we now also present our ShuTu measurements as average absolute values of the surface area of different neuronal compartments for both cell types, besides % differences (lines 149-150). We also note that we refer to the limitations of these datasets in Results (lines 151-152) and state in the Discussion that they lay the groundwork for future more detailed morphological studies (lines 487-490, 502-503).

3) Page 4 ln 179. Electrophysiology. The data is usefully presented in absolute terms in Table 1, but the differences should be described in the same terms, not as % (as in ln192) nor as qualitative comments (as in lns 198 – 203).

We agree with the reviewer’s request and now also present the differences in various parameters in terms of absolute average values, not only percentages (line 189-194). We have also eliminated the related qualitative comments that were originally at the end of the paragraph.

4) Page 5, ln 208-217. This is a methodological aside and is not relevant except to demonstrate the care that the authors have taken to exclude inhibitory neurons from their data set.

In agreement with the reviewer’s comment, we have now moved this section to Methods.

5) Page 5, ln 230-254. The data is consistent with a larger effect of TEA and 4-AP on RA repolarization. Again, absolute measures of duration rather than % change should be quoted and compared for the data in figure 4. This will be important if anyone were to model these conductances and neuronal properties in the future. APs in the Ald neurons are already longer, so a similar effect of TEA or 4AP might be expected to be proportionally smaller (if presented as %).

We thank the reviewer for this suggestion, as we are also interested in future modeling of these neuronal properties. We now present the absolute average measures with standard errors in addition to the % change in the description of the data shown in Figure 4 (lines 237-248). We also note that the individual paired data are shown in Figure 4—figure supplement 1.

Figure 4 clearly shows that TEA and 4-AP have direct effects on the AP of RA cells, but less so on Aid cells. But you can only infer that this is mediated by an action on Kv3 channels. The data is very difficult to interpret as all the numerical graphs are presented in fold change in AP halfwidth. How long were the drugs applied, and was recovery data collected to demonstrate washout? It is a pedantic point, but have you checked that the pharmacological evidence cited (from mammals) applies to birds? Since you show no effect in Supl Figure 4, a key issue would be: does α-DTx or iberiotoxin block Kv1 and BK channels (respectively) in the bird; have you positive controls or can you provide citations of such (or at least checked for precise homology at the respective toxin binding sites)?

As discussed above, we now present the absolute values in the text. We also include information on drug application times in the Methods (lines 894-897). Additionally, we now show wash out experiments with TEA, 4-AP and AUT5. The effects of TEA were almost completely reversed within 3 minutes of washout. We also observed a significant, but not complete washout with 4-AP on a significantly slower timescale than TEA. We did not observe a significant reversal of the effects of AUT5. This was not surprising, as a previous publication (Taskin et al., Eur. J. Pharmacol., 2015) has indicated that this compound does not washout readily. Nevertheless, these results confirm the stability of our recordings throughout the duration of our experiments. These data are now described in a new Figure 5—figure supplement 1, the figure legend, and the Results section (see lines 262-271, 442-444).

With regard to the effects of DTX and BK channels on neuronal excitability in birds, we note that both of these compounds have previously been shown to affect cellular properties in avian species. For example, administration of DTX in the nucleus magnocellularis in the chicken inhibits low threshold potassium currents (Rathouz and Trussell, J. Neurophysiol., 1998). Additionally, iberiotoxin inhibits a ca^2+^-dependent potassium current in the chicken ductus arteriosis (Moonen et al., Neonatology, 2010). We additionally performed amino acid sequence alignments for Kv1.1, Kv1.2, Kv7.2, Kv7.3 and BK channels between humans and zebra finches. We found that the percent identity was 93%, 98%, 77%, 84% and 92% respectively. Kv1.1 and 1.2 both contained high conservation (>95%) in the region between S5 and S6 implicated in the binding of α-DTX (Hurst et al., Mol. Phamacol. 1991, Tytgat et al., J. Biol. Chem., 1995). We additionally note a near 100% conservation of the S1-S6 region of Kv7.2 (97% identity) and Kv7.3 (99% identity) between zebra finches and humans. We also note that the BK channel showed remarkable conservation with 98% identity of the S0-S6 domains. Lastly, we note that there were detectable effects on evoked AP firing in RAPNs from administration of DTX. These effects appeared to be restricted to inter-spike periods and not the AP waveform, as shown in Figure 4—figure supplement 2A and Figure 5—figure supplement 2A. Thus, while this drug can exert detectable effects on finch RAPNs, we did not observe significant effects on their AP halfwidth. We now comment on these important aspects in lines 282-288.

6) The effect of TEA and 4AP differs in figure 5 (first AP) from figure 4 (AP train). The effect on the half-width of the 1st AP seems smaller than on subsequent APs in the train – particularly noticeable in 5A and 5B. What is the explanation for this (see point 8 below)?

We appreciate the reviewer’s comments. Regarding the first sentence, the examples shown in Figure 4 are spontaneous APs which were recorded with no current being injected. Figure 5 shows a train of APs evoked by a +500 pA current injection. Both RAPNs and AId neurons display broader APs, including the first AP, with the administration of TEA and 4-AP. The APs being shown in Figures 4 & 5 are on very different timescales, making visual comparisons of the halfwidth changes in these APs difficult to assess.

Regarding the second point, the increase in width of the subsequent APs in Figure 5 may be due to the accumulation of voltage-gated Na^+^ channels and/or A-type potassium channels in inactivated states. The decrease in Nav and/or Kv channel availability may add to the increase in the width of the subsequent APs in the train due to a relatively slower depolarization and/or repolarization rate. We note that this adaptation is described in our previous publication (Zemel et al., Nat. Commun., 2021), in which we show the first AP in a train (at +500 pA) is narrower, with significantly greater rates of depolarization and repolarization than subsequent APs. In order to acknowledge this adaption, we have changed our reference to RAPN and AId neuron firing as non-adapting to minimally adapting in the text (Lines 452,518).

7) The AP duration is also changing during the trains in Supl Figure 5. There are a few minor inconsistencies here. For example, XE991 has a significant effect in the lowest current injections in S5c. The control peak firing rates (spikes/sec) in S5E are lower (~70 vs ~100) than in S5A and S5C.

We appreciate again the reviewer’s comments, and refer the reviewer to the response to the previous comment to address the first sentence.

Regarding the second sentence, we acknowledge that there were indeed some changes in the evoked firing rates on APs in RAPNs in response to XE991. The statistical differences were shown in the legend of Figure 5—figure supplement 2. These findings in fact provide further supportive evidence that these drugs were exerting effects on neuronal cell excitability in finches, and suggest that Kv7 may contribute to the spike rates of RAPNs. Importantly, however, XE991 exerted no detectable effects on AP halfwidth or maximum rate of repolarization, as evidenced from the data from spontaneously firing APs in Figure 4—figure supplement 2. The difference in spontaneous firing rates of RAPNs exposed to XE991, while modest, may reflect the relative weight of the Kv7 conductance when cells are sparsely firing versus at more depolarized voltages during positive constant current injections, when additional voltage dependent conductances are more heavily recruited.

Regarding the third sentence, we acknowledge that on average the control peak firing rates are lower in Figure 5—figure supplement 2E compared to 2A and 2C. But these differences are very small, within the standard error for half of the current injection levels. Specifically, the averages for the controls in 2A and 2E were 35.2 ± 2.2 vs. 39.7 ± 4.3 at +100 pA, 53.8 ± 3.1 vs. 54.3 ± 5.6 at +200pA and 68.3 ± 4.1 vs. 64.5 ± 6.2 at +300pA. While the values for current injections above +300 pA were indeed lower the variability of the data in 2E was clearly greater, as indicated by the standard error. Taken together, while we cannot be certain as to what caused the decreased averages and higher variability in the two data points in Figure 5—figure supplement 2E, we do show that the RAPN evoked firing properties are remarkably consistent across experimental conditions (Figure 5, Figure 8, Figure 5—figure supplement 2).

8) Page 6 and figure 6. Voltage-clamp data is well presented and clearly shows larger outward currents in the RA neurons, with little or no 'low-voltage-activated' (steps to -30 mV; potential DTx-sensitive current) in either neuron type. The 'high-voltage activated' currents are very slow activating… And so would be hardly activated during a single fast AP lasting less than a ms….. This undermines your arguments that this is Kv3 on two counts: it is too slow to activate (taking over 100ms) and only differs by around 25% between RA and AId neurons. The key current for the AP repolarization, especially for the first AP in a train, is most likely to be the fast inactivating current. From the evidence in figure 6, it is not clear that this is a classic A-current (Kv4), because no estimation of steady-state inactivation has been conducted.Although this undermines your current hypothesis that this is a Kv3.1 current, the possibility that the Kv3 current is fast inactivating (i.e. a Kv3.3 or Kv3.4-like current) is very interesting. In Figure 6G, you appear to show that both the inactivating and slow-activating outward currents are TEA-sensitive. Because the activation of the slow outward current overlaps with the decay of the fast transient outward current, any estimation of one will be contaminated by the other. Ideally, you could steady-state inactivate the transient current so you can measure the slow-activating current. This would then allow you to estimate the 'contamination' of one current by the other.

We appreciate the thoughtful comments from the reviewer. We acknowledge that there is indeed a discrepancy between the time to the peak outward K^+^ current and the AP duration measured in RAPNs and AId neurons. There are a number of technical reasons that could explain this: (1) While we have identified conditions that maximize our ability to obtain high quality voltage-clamp recordings from RAPNs (see Zemel et al., Nat. Commun., 2021), we note that our cells likely retain some complex processes consisting of dendrites and local axon collaterals. This non-spherical geometry likely limits isopotentiality in our recordings, limiting our ability to accurately measure the speed of the onset of these currents. (2) While we successfully eliminated the vast majority of the Nav currents during our experiments using TTX, we note that our previous estimates of the Na^+^ current during the AP upstroke was ~65 nA (Zemel et al., Nat. Commun., 2021). We thus posit that a residual Na^+^ current that might remain even after administration of 1 µM TTX could limit our ability to fully isolate and measure the onset of the K^+^ current. (3) Under physiological conditions the outward K^+^ currents were too large to measure accurately. As stated in Methods, we lowered our intracellular K^+^ concentration by half in an attempt to limit the size of the outward current. This in turn reduces the current size, even at very early timepoints.

Nevertheless, we were able to reliably obtain outward K^+^ currents under the same experimental conditions from RAPNs and AId neurons. We note that the larger currents with shorter time to peak in RAPNs compared to AId neurons are consistent with a differential expression of Kv currents in these brain regions. Taken together with the differential expression of Kv3.1, but not other TEA-sensitive Kvs, our voltage clamp data provides further indication that Kv3.1 likely contributes to this difference between RAPNs and AId neurons.

We also agree with the reviewer that the overlap of any delayed rectifier current with the decay of the fast, transient outward current obscures any measurement of either of these components individually. The TEA sensitivity of these currents does suggest that Kv3 channels likely exist as heteromultimeric complexes, that include Kv3.4. This point is discussed in lines (584-588). While further studies on the components of the K^+^ current are of great interest to us, the data are consistent with a differential expression of a high-threshold, TEA-sensitive Kv channel between RAPNs and AId neurons.

9) P7 ln 335. Another paper has described kcnc3/Kv3.3 in a seabird (De Paoli-Iseppi et al. PLOS One 12: e0189181, 2017).

We thank the reviewer for pointing out this reference, which had previously escaped us. Contrary to the conclusions by De Paoli-Iseppi et al., however, we stand by our assertion that ours is the first description of the KCNC3/Kv3.3 in a bird. The basis for this statement is several-fold:

1) The locus identified in De Paoli-Iseppi et al. as the presumed fulmar KCNC3 ortholog turns out to be the fulmar KCNC4 ortholog, not KCNC3, as in fact acknowledged in their own S1 File, Table K. The accession number the authors provided (XM_009582050.1) is described in NCBI’s Refseq as Fulmarus glacialis potassium voltage-gated channel, Shaw-related subfamily, member 4 (KCNC4), transcript variant X1, mRNA. BLAST alignments of this transcript indeed reveal high hits only to KCNC4 in birds, sauropsids and mammals, including humans, thus this is clearly KCNC4, not KCNC3.

2) BLAST alignments of the primers (File S1, Table A in De Paoli-Iseppi et al) used to amplify the presumed CpG region of KCNC3 in the fulmar fully align to a scaffold (NW_009228155.1) that contains no trace of KCNC3, KCNC4, or any other voltage-gated ion channel, so the claim that these primers amplify a regulatory region of these genes cannot be verified;

3) Our study is the first to describe the KCNC3 genomic locus, including its synteny. Notably, the locus is on zebra finch chromosome 37, whose equivalent has not been assembled in either the fulmar or the shearwater, as well as in most other bird species. This hard to characterize microchromosome has only been completed in very few avian species, requiring the latest advances in long-read sequencing and assembling technologies (Rhie et al., 2021), which helps to explain why this remains an undefined locus in the majority of bird species.

4) There are no significant hits to a KCNC3-like locus in fulmar or shearwater in our unbiased BLAST searches of genomic databases using as queries the identified KCNC3 in zebra finch, gyrfalcon and kakapo. Altogether, we conclude that the locus mentioned as KCNC3 was erroneously identified as KCNC3 and most likely is KCNC4. We have added an explanatory comment with regard to this misannotation in the Results. We also provide further details of our curation/annotation efforts in Methods, as well as include in the legend of Figure 7—figure supplement 1 the loci numbers for all avian genes that were misannotated in NCBI and that we have reannotated as part of our curation effort.

10) Figure 7 and Page 7/8. The heading on ln 319 states, that Kv3.1 is the TEA-sensitive ion channel subunit… But this seems a rather weak point and could be disputed, as you have measured mRNA, not protein (and the interpretation of the current is ambiguous). The in situ hybridisation for each of the Kv3 genes is interesting, but the quantification is insufficient. The lack of absolute values (or at least relative to some 'housekeeping' gene) for the mRNA expression is a serious weakness. The ratios expressed here between two nuclei in Figure 7 hide/obscure important information and have contributed to the misinterpretation of the physiology. For example, one could argue on the basis of Fig7A that all Kv3 genes are present and that kcnc3 is most striking in that it highlights the two nuclei of interest, is present in both and you have the positive control of Purkinje neurons. Fig7B and 7C are best omitted… The expression ratio of RA to Ald has little or no relevance to a measurable parameter in your study. Perhaps you could argue some relevance if you had done a more quantitative measure and/or western blotting, but this has not been done. Your own evidence clearly indicates that multiple Kv3 subunits are present (Figure 7A), but you have no measure of the efficacy with which the mRNA is translated into protein (it is also hard to know what constitutes a 'background' value for any of the mRNA probes).

The reviewer brings up important points with regard to mRNA expression of Kv3 subunits. In retrospect, we agree that the heading of that section may have been somewhat misleading, and have adjusted it accordingly. It was not our intention to claim that Kv3.1 is the sole channel in this family that is expressed in RAPNs, or that it is the sole determinant of RAPN properties, nor was our intention to disprove the possible contributions of other Kv3 family subunits to RAPN properties. Our goal was to demonstrate that Kv3.1 differential expression in RAPNs vs AId neurons in a pattern consistent with the regional differences in their electrophysiological properties, thus indicative of Kv3.1 playing a larger role in RAPNs compared to AId neurons. In that regard, we disagree that our analysis is misleading or irrelevant. We do show that other Kv3s are expressed in RAPNs, but we are saying that their mRNA expression patterns are inconsistent with the differences seen between RAPNs and AId spikes. Thus, these other family members are less likely to contribute to the differences in electrophysiological properties seen between RAPNs and AId neurons. We agree that the Kv3.3 pattern is interesting, suggesting a prominent role of this subunit in both RAPNs and AId neurons, but the high Kv3.3 expression in both areas is inconsistent with the differential electrophysiological properties seen between these two regions. We have modified the relevant passages of the text to make these points more clearly (see lines 85-87, 105-116, 395-399, 409-410, 446, 546, 551-553, 1592, 1624).

To further address the reviewer’s concerns, and for disclosure and completeness, we note that the absolute values of our densitometric analysis that were used to generate the ratios are provided in the corresponding tab 7B and 7C of the Source Data file that accompanies the manuscript, as is also done for electrophysiological parameters. We also note that the in situ signal is sensitive to differences in probe length, hybridization strength and stringency conditions, and does not involve counts of mRNA molecules, thus the values do not allow us to conclude which specific subunit is most highly expressed in RAPNs, which again was not our goal. We closely examine probe specificity through genomic alignment of probe sequences, and our in situs are run under optimized hybridization conditions that result in no detectable signal upon omission of probe or anti-dig antibody. Background levels as measured in tissue areas devoid of cells are well below the cellular levels measured for each of the probes used, as attested in the Source Data file. Furthermore, the large number of probes in Figures 7B/C that show no differences between regions essentially play a normalizing role (similar to the request for “housekeeping” gene data), showing that transcript of various lengths and sequences are not differential between RAPNs and AId neurons, and further helping to establish the specificity of the differential expression of Kv3.1 between RAPNs and AId. Lastly, we agree that we cannot directly infer protein levels from mRNA expression, and note that there is currently no published use of Kv3.1 antibodies in the zebra finch. However, based on parsimony, we think it is reasonable to infer that the Kv3.1 mRNA pattern correlates better with the electrophysiological differences between RAPNs and AId neurons than any of the other subunits examined. We have edited several passages in Methods and Results to address these important points.

11. AUT5. This is not a well-designed experiment, since you apply a dose that is half-maximal (1uM when the EC50 is 1.3uM: ln 391), so you would be unlikely to observe more than half of the potential effect. For most readers, this comes across as a rather minor point and I think they will wonder why they need to know this information, certainly it is a weak figure with which to finish. Perhaps if AUT5 was shown to change the finch song or other behaviour in the opposite way to blocking the channel it would be interesting. AUT5 effects have also been noted previously in other publications:2015. Biophysical characterization of KV3.1 potassium channel activating compounds. Taskin B, von Schoubye NL, Sheykhzade M, Bastlund JF, Grunnet M, Jespersen T.Eur J Pharmacol. 2758:164-70.2017. Kv3.1/Kv3.2 channel positive modulators enable faster activating kinetics and increase firing frequency in fast-spiking GABAergic interneurons. Boddum K, Hougaard C, Xiao-Ying Lin J, von Schoubye NL, Jensen HS, Grunnet M, Jespersen T. Neuropharmacology, 118:102-112.

We appreciate the reviewer’s comments and note that we have cited both of these studies within our manuscript. We additionally note that the half maximal dose of AUT5 yielded significant effects on RAPNs, but not on AId neurons. These results are consistent with the difference in expression of Kv3.1 between RAPNs and AId neurons. We believe that the AUT5 data is critical in: (1) Narrowing down to Kv3.1/Kv3.2 as the most likely contributing channels for the differences between RAPNs and AId neurons; and (2) Showing that by positively modulating Kv3.1 channel activity we can demonstrate opposite effects of blockade by TEA or 4-AP. With regard to delivering AUT5 to RA during song production, it would require various elaborate controls and the exact predictions are unclear, as one might see opposing effects due to AUT5’s possible actions on Kv3.1/2 expressed in both RAPNs and GABAergic interneurons. Thus, while we agree this is a tantalizing experiment, we believe it is beyond the scope of the present study.

Reviewer #3 (Recommendations for the authors):This paper was a pleasure to read and goes through a logical sequence of experiments to pinpoint the role of Kv3(.1?) in the fast-spiking of RAPN neurons. There are large replicate numbers for almost all findings, and admirably, there is confirmation of the main result at physiological temperature. The phase plots in Figure 4 make clear that 4AP and TEA, at the concentrations used, have little effect on Ald neurons, but strong effects on RAPNs in both current clamp and voltage clamp. The methods developed to improve voltage clamp compliance are to be commended.1) The paper has an unclear focus. In some ways, it focuses on the properties of RAPN neurons and how they might generate fast APs. Yet in other contexts, the paper seems to center on the description of a previously uncharacterized cell type, the neurons of Ald.

We appreciate the reviewer’s concern. Our main goal has been to characterize the excitable properties of RAPNs, which are upper motor neurons with a specialized role in vocal-motor control. However, we also discovered that the excitable properties of RAPNs differ in significant ways from those of neurons in the adjacent AId, which are thought to control non-vocal somatic motor function (Mandelblat-Cerf, *eLife*, 2014; Yaun and Bottjer, eNeuro, 2020; Feenders et al., PLoS One, 2008). We then realized that an important way to highlight the unique properties of RAPNs would be to contrast them with those of AId neurons. Whereas there is literature describing the excitable properties of RAPNs, no studies to date have described the intrinsic excitable properties of AId neurons (using whole-cell patch clamp), nor contrasted the neuronal excitable properties of these two regions. Examination of the differences between RAPNs and AId neurons provided key insights into specializations that may be associated with vocal production. To clarify this important point, we now make the goals and general strategy of our study more explicit in the beginning of the Results section (see lines 105-116).

2) The pharmacological analysis of repetitive spike firing is lacking. For example, the traces in Figure 5B and 5D, and 8B and 8D, DO in fact show effects of 4AP/TEA/AUT5 on Ald neurons. This raised two issues. A) Repeated measures, such as a paired t-test I (before and after the drug, on a per cell basis) are arguably more appropriate than group averages at each current injection strength. B) Some effort should be dedicated to reversibility. While it can be technically challenging to hold a cell for the total time required to completely wash out the drug, at least a few examples should be provided to confirm that "drug" effects are not due to time-dependent changes in cell physiology.

We appreciate the comment about the need for statistical rigor and drug reversibility experiments. With regard to the first point, we did run repeated measures statistics. In our experimental design we measured spikes per second at multiple levels of positive current injections before and after drug administration. Based on this experimental design, we conducted a Repeated Measures Two-way ANOVA to determine any effects on the spikes per second of the recorded cells, which is statistically more appropriate than running multiple t-tests. For datasets in which the ANOVA detected a significant effect, we followed up with post-hoc tests to determine significance for individual comparisons that could explain our results. While we note the trends in the data for AId neuron responses to 4-AP and TEA (see lines 252-260), no significant effect was detected by the ANOVA. To our knowledge we used an appropriate and rigorous statistical treatment of the data given the experimental design.

Per the reviewers’ suggestion (see also Reviewer 1) we performed washout experiment with TEA, 4-AP and AUT5. These data are now described in Results (lines 262-271, 442-444) and presented in the new Figure 5—figure supplement 1.

3) For fast repolarization, it would likely be better to focus mainly on the early current, highlight with the red bars in figure 6, rather than on the current persisting until 200 ms.

We agree that the early current indeed plays a more important role in AP repolarization. While we analyzed the currents at 200 ms, we gave equal attention to analysis of the initial inactivating current observed. We have now added a sentence to the discussion focusing on the early current and its putative effects on the repolarization of APs in RAPNs and AId neurons. See lines (588-589).

4) There is a bit of circular logic regarding the description of the effects of AUT5 on spike depolarization (Figure 8A). If RA affects the maximum depolarization, then this could secondarily increase the recruitment of K channels and in turn produce a faster rate of repolarization.

The reviewer comments here on another important point. We note, however, that there was no change in the maximum rate of spike depolarization in RAPNs in response to AUT5 (stats in lines 425-426). While we agree that Kv channel recruitment could result from a more depolarized spike peak, as far as we are aware there is no data to suggest effects of AUT5 on Nav channels, that could cause this increased peak. Previously published literature (Boddum et al., Neuropharmacol., 2017; Taskin et al., Eur. J. Pharmacol., 2015) and personal communications with Autifony Therapeutics describe AUT5 as being a highly selective positive modulator for Kv3.1 and Kv3.2. We acknowledge that the exact explanation for the effects of AUT5 on spike shape may be complicated, with indirect effects on the availability of other voltage gated ion channels during spiking as a result of voltage-dependent shifts, activation changes and deactivation changes of Kv3.1 channels in response to 1 µM AUT5. Nonetheless, coupled with the other AUT5-induced changes that were seen in RAPNs, but not in AId neurons, we believe that the most parsimonious explanation for our data is that the changes seen in RAPNs with AUT5 are due to the positive modulation of the Kv3.1 subunit.

We have now included a statement regarding the potential increase in recruitment of Kv’s as a result of the depolarized peak of the AP (See lines 432-433).

5) While voltage clamp compliance is likely much improved by the approaches taken, it will still not be perfect, and there will remain issues of non-isopotentiality. This should be discussed.

We appreciate the reviewer’s comments and revised text in Results to address issues of isopotentiality. See lines 295-301.

6) The correspondence to Betz cells is oversold and should be removed from the title and abstract. There are many differences, as the authors note. A) The RA cells are not necessarily larger than other motor output neurons as with Betz cells, B) They have presumably developed along a unique evolutionary path, C) Other central features of Betz cells (conduction velocities, myelination, axon fiber diameter) are not reported.

We appreciate the reviewer’s comments and have removed the reference to Betz cells from the title. We have additionally modified the Abstract and significantly changed the Introduction (eliminating the first paragraph) to de-emphasize the comparison with Betz cells. While we also agree that future studies are indeed required (i.e. looking at myelination, axon fiber diameter and conduction velocities) for thorough comparisons, we nonetheless believe it is worth discussing the similarities to Betz cells in the paper. In particular, the presence of Kv3.1b in Betz cells (but not in other Layer 5 motor cortex pyramidal neurons) has been highlighted by researchers working with primates, so we find it most intriguing that this important feature is also present in the RA cells. Nevertheless, the comparison with Betz cells is now mostly detailed in the Discussion section.